# Pushing the Limits of Block Rotations in Post-Training Quantization

Sai Sanjeet [* 1 2]   Ian Colbert [* 1]   Pablo Monteagudo-Lago [1]   Giuseppe Franco [1]
Yaman Umuroglu [1 3]   Nicholas J. Fraser [1]

## Abstract

Recent post-training quantization (PTQ) methods have adopted block rotations to diffuse outliers prior to rounding. While this reduces the overhead of online full-vector rotations, the effect of block structure on outlier suppression remains poorly understood. To fill this gap, we present the first systematic, non-asymptotic analysis of outlier suppression for block Hadamard rotations. Our analysis reveals that outlier suppression is fundamentally limited by the geometry of the input vector. In particular, in the deterministic worst case, post-rotation outliers are minimized when the pre-rotation $\ell_1$ norm mass is evenly distributed across blocks. Guided by these insights, we introduce PeRQ (**Pe**rmute, **R**otate, then **Q**uantize), a PTQ framework that redistributes activation mass via permutations prior to rotation. We propose a greedy mass diffusion algorithm to calibrate permutations by equalizing the expected blockwise $\ell_1$ norms. To avoid adding inference overhead, we identify permutation-equivariant regions in transformer architectures to merge these permutations into model weights before deployment. Experiments show that PeRQ consistently improves accuracy across all block sizes, recovering up to 90% of the full-vector rotation perplexity when quantizing Llama3 1B to INT4 with block size 16, compared to 46% without permutations.

## 1. Introduction

Post-training quantization (PTQ) remains essential for improving the efficiency of large language models (LLMs), yet activation outliers continue to limit model accuracy in few-bit settings. Recent methods insert rotations into the compute graph to diffuse these outliers across vector coordinates, often substantially improving accuracy. However, while some rotations can be merged into model weights before deployment, others must be computed online during inference, incurring non-trivial runtime overheads.

A natural approach to reduce the cost of online rotations is to partition activation vectors into fixed-sized blocks and apply rotations independently within each block. When the rotations are Hadamard matrices, this reduces compute requirements from $\mathcal{O}(d \log d)$ to $\mathcal{O}(d \log b)$ for a $d$-dimensional activation vector and block size $b$. In practice, this can yield meaningful runtime improvements: Shao et al. 2025 report a $1.5\times$ reduction in online rotation overhead for Llama2 7B running at MXFP4 using block Hadamard rotations with $b = 32$, compared to full-vector Hadamard rotations, corresponding to a 2% reduction in end-to-end latency. However, this efficiency gain is not free. We prove that reducing block size $b$ can increase worst-case post-rotation outliers with high probability, later formalized in Proposition 3.4. Figure 1 illustrates this effect within Llama3 1B: while full-vector rotations substantially reduce activation ranges, smaller block rotations are often less effective at suppressing outliers. As a result, block rotations can introduce a trade-off between computational efficiency and model accuracy for quantized LLMs. In this work, we identify and mitigate fundamental limitations of block Hadamard rotations to improve outlier suppression while retaining efficiency.

**Contributions.** We introduce PeRQ, a new framework for calibrating permutations to improve the outlier suppression capabilities of block rotations for few-bit LLM inference. Our work is driven by three key contributions. **First,** we present the first non-asymptotic analysis of the outlier suppression capabilities of full-vector and block Hadamard rotations. Prior work asymptotically analyzes expected $\ell_2$ error after block rotations (Egiazarian et al., 2026), offering insights under Laplacian or Gaussian assumptions but no worst-case guarantees. In contrast, our analysis yields the first deterministic bounds that provide sufficient conditions under which full-vector and block Hadamard rotations suppress outliers (Propositions 3.1 and 3.2). Our analysis reveals that outlier suppression is fundamentally limited by the geometry of the input vector, namely its $\ell_1$ norm mass distribution. Moreover, we analyze how these limits evolve

---

*Equal contribution   [1]Advanced Micro Devices, Inc. (AMD) [2]State University of New York at Buffalo [3]Norwegian University of Science and Technology. Correspondence to: Ian Colbert <ian.colbert@amd.com>.

*Proceedings of the 43rd International Conference on Machine Learning*, Seoul, South Korea. PMLR 306, 2026. Copyright 2026 by the author(s).

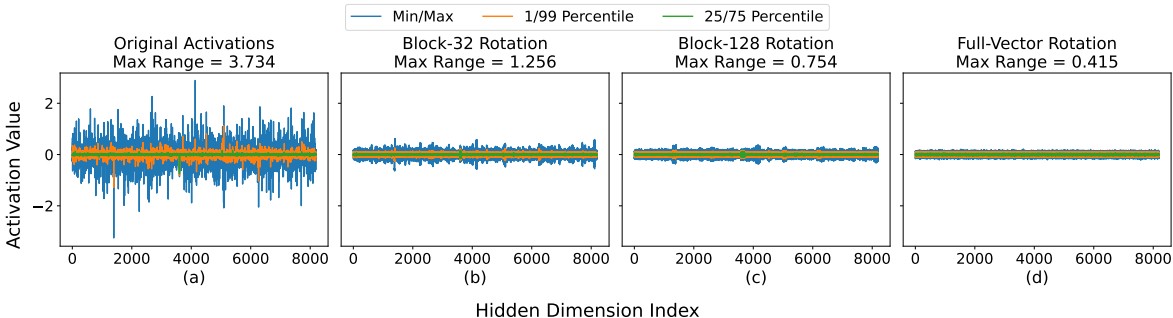

*Figure 1.* We visualize input activation distributions sampled from 2048 tokens of WikiText2 at the third down projection layer in Llama3 1B under four configurations: (a) original model, (b) block Hadamard rotation with $b = 32$, (c) block Hadamard rotation with $b = 128$, and (d) full-vector rotation. As $b \to d$, the activation range decreases, showing block rotations can reduce outlier suppression effectiveness.

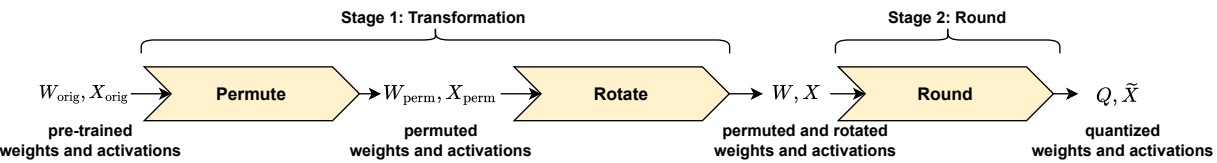

*Figure 2.* Given pre-trained weights and activations, PeRQ first computes a permutation that equalizes the per-block activation mass. The permuted weights and activations are then rotated. Finally, the permuted and rotated weights and activations are then rounded to a target alphabet defined by a bit width, scaling factor, and zero point.

with block size, both deterministically (Corollary 3.3) and probabilistically (Proposition 3.4). **Second,** we propose a greedy mass diffusion algorithm (MassDiff) that uses calibration data to construct permutation matrices to equalize full-precision activation mass across blocks. **Third,** we show that permutations can be merged into surrounding layers by identifying and exploiting permutation-equivariant regions in neural network architectures, yielding an end-to-end framework that improves the outlier suppression capabilities of block Hadamard rotations without introducing additional inference-time overhead. In Section 5, we show that PeRQ improves the accuracy of few-bit models compared to existing baselines, completely closing the gap to full-vector rotations with block sizes of 128 or higher.

**Notation.** Throughout, input activation $X \in \mathbb{R}^d$ is a row vector, where $d = nb$ for $n$ blocks of $b$ elements. The $j$-th block of $X$ is denoted $X_{\{j\}} \in \mathbb{R}^b$ for $j \in [n]$, where $[n] = \{1, \dots, n\}$. $R \in \mathbb{R}^{k \times k}$ denotes a normalized Hadamard matrix, where $R_i \in \mathbb{R}^k$ is the $i$-th column of $R$ with $\|R_i\|_2 = 1$ and $\|R_i\|_\infty = 1/\sqrt{k}$ for $i \in [k]$. A block rotation is a block-diagonal matrix constructed via the Kronecker product $\tilde{R} = I_n \otimes R = \mathrm{diag}(R, \dots, R)$, where $I_n \in \mathbb{R}^{n \times n}$ is the identity matrix. Note that $\tilde{R} \in \mathbb{R}^{d \times d}$ when $k = b$. $P \in \mathbb{R}^{d \times d}$ is a permutation matrix if there exists a permutation $\pi : [d] \to [d]$ such that $Pe_i = e_{\pi(i)}$ for all $i \in [d]$. A quantizer $\mathcal{Q} : \mathbb{R} \to \mathcal{A}$ maps real values to a discrete $q$-bit alphabet $\mathcal{A} \subset s\mathbb{Z}$, where $s$ is the step size (or resolution) and $|\mathcal{A}| \le 2^q$. We specify the quantizers used for the data formats in this work in Appendix B.

## 2. Background and Related Work

Recent advances in PTQ can generally be organized into two stages of a quantization pipeline, as illustrated in Figure 2. The first involves applying transformations to the full-precision weights or activations. The second involves mapping the resulting weights or activations to a discrete target alphabet (*e.g.*, 4-bit integers), typically via error-correcting rounding algorithms. Our work focuses on the first stage.

**Transformation algorithms.** Depending on how transformations are incorporated into the compute graph, they may either be merged into model tensors as an offline preprocessing step before deployment (*mergeable*), or remain explicitly in the graph to be evaluated at inference time (*online*). One line of work equalizes dynamic range with channelwise scaling prior to rounding, such as SmoothQuant (Xiao et al., 2023) and AWQ (Lin et al., 2024b). Another line of work inserts rotations into the graph prior to rounding to suppress outliers in both the weights and activations. These rotations may be random (Ashkboos et al., 2024), expandable (Franco et al., 2025a), or learned (Liu et al., 2025). While mergeable rotations can be applied without impacting inference-time computation, online rotations incur non-trivial inference overhead (see Appendix A for a full discussion). This cost motivates the study of structured rotation schemes that trade expressivity for computational efficiency while preserving outlier suppression benefits.

**Structured rotations.** Early works with online rotations adopt dense matrices (Chee et al., 2023), requiring $\mathcal{O}(d^2)$

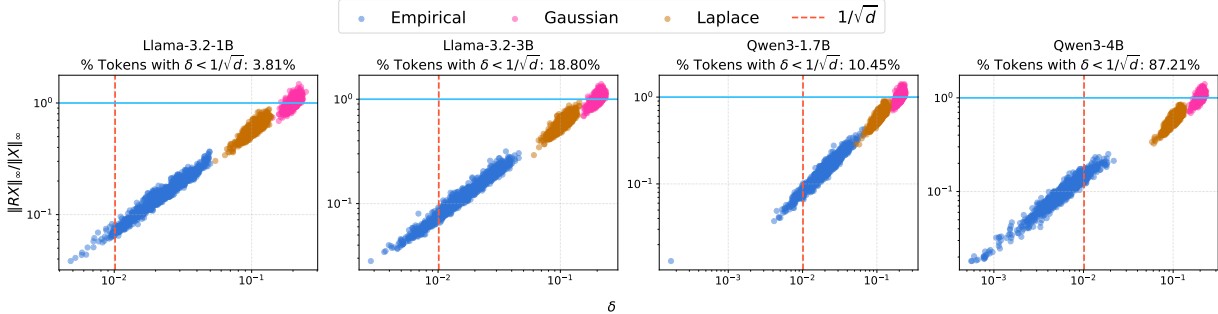

*Figure 3.* In **blue**, we show values of mass concentration $\delta$ and the outlier suppression ratio $\|XR\|_\infty/\|X\|_\infty$ for 1024 tokens sampled from the third down projection layer of Llama3 and Qwen3 models, using activations from WikiText2. The reference value $1/\sqrt{d}$ is shown in **red**, below which Proposition 3.1 guarantees outlier suppression. Although many tokens lie above this sufficient threshold, outlier suppression is observed consistently, and its degree is strongly correlated with $\delta$. For comparison, Gaussian and Laplacian distributions are first fit to the empirical activations on a per-token basis, and $\delta$ values are then computed from samples drawn from these fitted distributions; their resulting $\delta$ distributions differ substantially from those observed for real LLM activations, yet the correlation persists.

operations to rotate $X \in \mathbb{R}^d$. One line of work reduces this complexity by imposing algebraic structure on full-vector rotations. For example, FlatQuant (Sun et al., 2024) learns Kronecker decompositions of dense matrices, reducing the cost to $\mathcal{O}(d\sqrt{d})$. Other approaches further restrict structure: QuaRot adopts random Hadamard rotations and ButterflyQuant (Xu et al., 2025) learns factorizations of Givens matrices, both admitting $\mathcal{O}(d\log d)$ implementations. A complementary strategy is to independently apply rotations to fixed-size partitions of $X$, yielding block rotations. DuQuant (Lin et al., 2024a) composes dense block rotations with permutations, reducing memory to $\mathcal{O}(b^2)$ for block size $b$ while requiring $\mathcal{O}(db)$ operations at inference. In practice, block Hadamard rotations have emerged as the most common structured rotation, reducing the computational cost to $\mathcal{O}(d\log b)$. Recent concurrent works such as MR-GPTQ (Egiazarian et al., 2026) and BRQ (Shao et al., 2025) study block Hadamard rotations with microscaled (MX) datatypes, underscoring their practical relevance in modern PTQ pipelines. Our work takes a complementary perspective. Rather than proposing a new family of structured rotations or focusing on their interaction with specific numerical formats, we analyze the outlier suppression capabilities of Hadamard rotations, and use the resulting insights to design an equalization framework that maintains these capabilities under block structure constraints.

## 3. Theoretical Analysis of Outlier Suppression

The connection between outlier suppression and quantization error can be seen through deterministic worst-case bounds. For integer quantizer $\mathcal{Q}$, the step size (or scaling factor) is commonly defined as $s = \|X\|_\infty/(2^{q-1}-1)$ for a $q$-bit alphabet (Gholami et al., 2022). The worst-case quantization error for activation vector $X \in \mathbb{R}^d$ satisfies

$$\|X - \mathcal{Q}(X)\|_2 \le \frac{\sqrt{d}}{2^q - 2}\|X\|_\infty.$$

Indeed, the worst-case quantization error scales linearly with $\|X\|_\infty$, so suppressing outliers (*i.e.*, reducing $\|X\|_\infty$) directly tightens worst-case $\ell_2$ error.

Using this lens, we first establish sufficient conditions under which full-vector Hadamard rotations suppress outliers (Proposition 3.1). We then generalize our analysis to block Hadamard rotations (Proposition 3.2), and characterize how worst-case post-rotation outliers evolve with block size both deterministically (Corollary 3.3) and probabilistically (Proposition 3.4). All proofs are provided in Appendix D.

### 3.1. Full-Vector Hadamard Rotations

We first offer a deterministic and non-asymptotic analysis of when full-vector Hadamard rotations suppress outliers in activation vector $X$ with the following proposition.

**Proposition 3.1** (When Full-Vector Hadamard Rotations Suppress Outliers)**.** *Let $X \in \mathbb{R}^d$ be an activation vector, and let $R \in \mathbb{R}^{d \times d}$ be a normalized Hadamard matrix. Define $\delta = \|X\|_1/(d\|X\|_\infty)$, then the rotation of $X$ by $R$ satisfies*

$$\|XR\|_\infty \le \delta\sqrt{d}\|X\|_\infty. \tag{1}$$

Intuitively, the degree of outlier suppression varies with the geometry of $X$ as captured by $\delta$, which can be interpreted as the concentration of mass in $X$. Since $\|X\|_\infty \le \|X\|_1 \le d\|X\|_\infty$, we have $\delta \in [d^{-1}, 1]$. Larger values of $\delta$ imply near-uniform magnitude, whereas smaller values of $\delta$ imply mass concentration in a few large outliers.

Our bound also yields a clear guarantee: $\|XR\|_\infty < \|X\|_\infty$ when $\delta < 1/\sqrt{d}$. Importantly, this condition is sufficient but not necessary; outlier suppression can and does occur in practice for activation vectors with $\delta$ well above this threshold. Figure 3 visualizes the empirical behavior of $\delta$ for Llama3 and Qwen3 models, showing that outliers are consistently suppressed across all sampled tokens even when the sufficient condition is not satisfied. Moreover, Figure 3

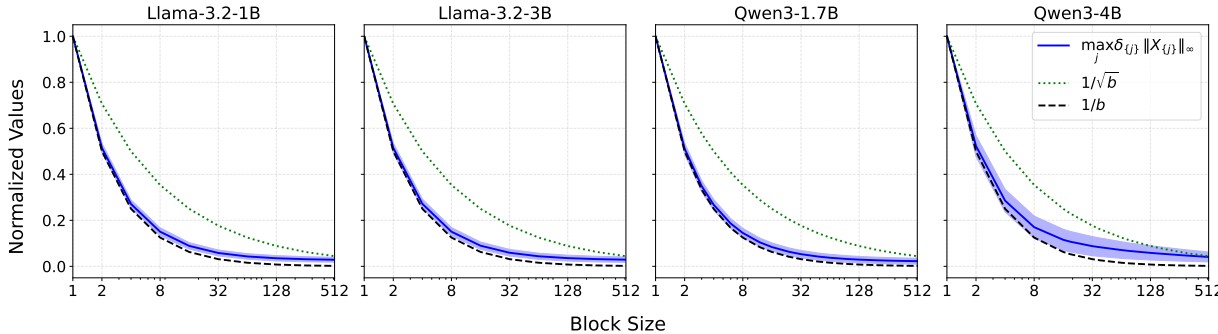

*Figure 4.* In **blue**, we plot, for each block size $b$, the empirical mean and standard deviation of $\max_j \delta_{\{j\}}\|X_{\{j\}}\|_\infty$ normalized by $\|X\|_\infty$. Following Proposition 3.2, we show the sufficient threshold for outlier suppression $1/\sqrt{b}$ in **green**, and the theoretical lower bound $1/b$ in **black**. Values are computed over all down projection layers using 10K tokens sampled from the WikiText2 dataset.

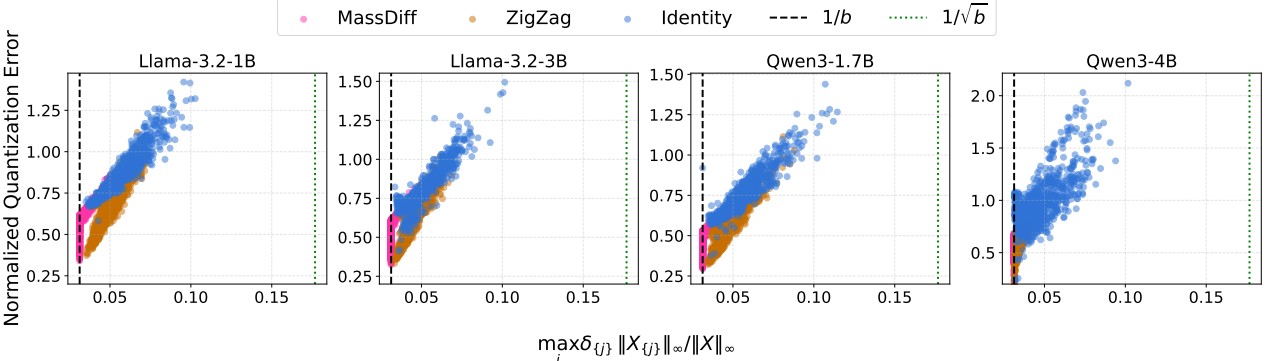

*Figure 5.* For each token in Figure 3, we compute (1) $\max_j \delta_{\{j\}}\|X_{\{j\}}\|_\infty$, which is the bound from Proposition 3.2 divided by a constant $\sqrt{b}$, and (2) the actual quantization error, defined as $\|XP\tilde{R} - Q(XP\tilde{R})\|_F$, after 4-bit quantization with block size $b = 32$ (both are normalized by $\|X\|_\infty$ per token). Consistent with Figure 4, we show the sufficient threshold for outlier suppression $1/\sqrt{b}$ in **green**, and the theoretical lower bound $1/b$ in **black**. In **blue**, we show the values without a permutation (*i.e.*, Identity $P = I$). We then compare **MassDiff** (Algorithm 1) and **ZigZag** (Lin et al., 2024a), where permutations are calculated per-token. The resulting scatter plot reveals that our normalized bound closely tracks relative per-token quantization error. Moreover, MassDiff reduces the bound for 100% of tokens across all four models, with mean quantization error reductions of 37.5–40.5%. In contrast, ZigZag only reduces the bound on 82–90% of tokens with 21–36% error reduction, providing strong evidence that our theory identifies the correct bottleneck: mass concentration.

provides two additional insights. First, the empirical distribution of $\delta$ for real LLM activations differs markedly from those implied by common distributional assumptions. To ensure a fair comparison, Gaussian and Laplacian models are fit to the empirical activations on a per-token basis, and $\delta$ values are then computed from samples drawn from these fitted distributions. The resulting distributions fail to capture the empirical distribution of $\delta$ observed in real models, highlighting the limits of analyses based on such assumptions. Second, $\delta$ is strongly correlated with the outlier suppression ratio $\|XR\|_\infty/\|X\|_\infty$, indicating that mass concentration serves as an effective proxy for predicting the degree of outlier suppression achieved by rotation.

### 3.2. Block Hadamard Rotations

Restricting rotations to operate on fixed-size partitions of $X$ fundamentally changes how mass is redistributed across coordinates, as formalized by the following proposition.

**Proposition 3.2** (When Block Hadamard Rotations Sup-

press Outliers). *Let $\tilde{R} = I_n \otimes R$ be a block rotation, where $R \in \mathbb{R}^{b \times b}$ is a normalized Hadamard matrix and $\tilde{R} \in \mathbb{R}^{d \times d}$ with $d = nb$. Let $X_{\{j\}} \in \mathbb{R}^b$ be the $j$-th block of $X \in \mathbb{R}^d$ for $j \in [n]$. Define $\delta_{\{j\}} = \|X_{\{j\}}\|_1 / (b\|X_{\{j\}}\|_\infty)$, then the rotation of $X$ by $\tilde{R}$ satisfies*

$$\|X\tilde{R}\|_\infty \leq \max_{j \in [n]} \delta_{\{j\}}\sqrt{b}\,\|X_{\{j\}}\|_\infty. \quad (2)$$

Proposition 3.2 provides a principled characterization of when block Hadamard rotations suppress outliers. Importantly, it strictly subsumes Proposition 3.1; Equation 2 reduces exactly to Equation 1 as $b \to d$. Moreover, since $\delta_{\{j\}}\|X_{\{j\}}\|_\infty = \|X_{\{j\}}\|_1/b$, outlier suppression in the block setting is fundamentally limited by the local geometry of the activation vector, namely the block with the largest $\ell_1$ norm mass. This dependence is captured by the per-block ratio $\delta_{\{j\}} \in [b^{-1}, 1]$, which plays the same role as $\delta$ in the full-vector case but is now tied to both the block size $b$ and the local geometry within each block. For fixed block size $b$, larger values of $\delta_{\{j\}}$ again correspond to blocks whose

coordinates are more evenly distributed. Therefore, the bound in Equation 2 is dominated by blocks that exhibit both (1) high intra-block uniformity (large $\delta_{\{j\}}$) and (2) large $\|X_{\{j\}}\|_\infty$ relative to $\|X\|_\infty$. For fixed activation vector $X$, one can reason mechanistically about how outlier suppression evolves with $b$ via the following corollary.

**Corollary 3.3** (Deterministic Evolution of Post-Rotation Outliers)**.** *Let $X \in \mathbb{R}^d$ be an activation vector and let $X_{\{j\}} \in \mathbb{R}^b$ be the $j$-th block of $X$ for $j \in [n]$ with $d = nb$. Define $\mathcal{Z}(b; X) = \max_{j \in [n]} \sqrt{b}\, \delta_{\{j\}}\|X_{\{j\}}\|_\infty$, where $\delta_{\{j\}} = \|X_{\{j\}}\|_1 / \left(b\|X_{\{j\}}\|_\infty\right)$. Then, for positive integers $k, b' \in \mathbb{N}$ such that $b = kb'$, it is verified that*

$$\mathcal{Z}(b; X) \leq \sqrt{k}\, \mathcal{Z}(b'; X).$$

Given $X$ and atomic block size $b'$, increasing the block size to $b = kb'$ causes the worst-case post-rotation outlier to grow by a factor of $\sqrt{k}$. This arises from the inherent pessimism of worst-case analysis: larger $b$ increases the number of terms contributing to each rotated coordinate, and these terms can, in the worst case, align constructively with the Hadamard matrix. Although this deterministic view is informative, it does not directly model expected behavior, since the interaction between the block structure of $\tilde{R}$ and the local geometry of $X$ is non-trivial.

To better understand the expected behavior of Equation 2, we visualize the empirical relationship between $\max_j \delta_{\{j\}}\|X_{\{j\}}\|_\infty/\|X\|_\infty$ and block size $b$ in Figure 4. By definition, $\delta_{\{j\}} \geq b^{-1}$, so this quantity is lower bounded by $1/b$. Moreover, Proposition 3.2 also yields a sufficient condition for guaranteed outlier suppression: $\|X\tilde{R}\|_\infty < \|X\|_\infty$ when $\max_j \delta_{\{j\}}\|X_{\{j\}}\|_\infty/\|X\|_\infty < 1/\sqrt{b}$. Across down projection layers in Llama3 and Qwen3 models, the observed values of $\max_j \delta_{\{j\}}\|X_{\{j\}}\|_\infty/\|X\|_\infty$ fall well-below this threshold for a wide range of block sizes. Thus, Proposition 3.2 predicts outlier suppression in this regime. Moreover, the normalized bound closely tracks relative per-token quantization error in practice (Figure 5), serving as a concrete target for the algorithm design discussed in Section 4. To complement this observation analytically, we provide a probabilistic analysis of how post-rotation outliers evolve with block size via the following proposition.

**Proposition 3.4** (Probabilistic Evolution of Post-Rotation Outliers)**.** *Let $\tilde{R} = I_n \otimes R$ be a block rotation, where $R \in \mathbb{R}^{b \times b}$ is a normalized Hadamard matrix and $\tilde{R} \in \mathbb{R}^{d \times d}$ with $d = nb$. Given $Y = (Y_1, \ldots, Y_d) \in \mathbb{R}^d$, let $S = (S_1, \ldots, S_d)$ be a random vector with i.i.d. Rademacher entries $S_i \sim \mathrm{Rad}(\pm 1)$. Define $X \in \mathbb{R}^d$ coordinate-wise as $X_i = S_i Y_i$ for $i \in [d]$. Then, conditional on $Y$,*

$$\|X\tilde{R}\|_\infty \leq \sqrt{\frac{2}{b} \log\left(\frac{2d}{\varepsilon}\right)\|X\|_2^2} \qquad (3)$$

*with probability at least $1 - \varepsilon$.*

From Equation 3, we can reason about the expected behavior of post-rotation outliers beyond the adversarial worst case. Under mild assumptions, which we test in Appendix D.4, the expected worst-case post-rotation outlier decreases with block size $b$ with high probability. This reveals an explicit trade-off: larger blocks decrease maximum post-rotation outliers with high probability but increase minimum computational requirements (Appendix A). Moreover, Proposition 3.4 offers a complementary geometric perspective. In particular, outlier suppression is also probabilistically limited by mass concentration (see Remark D.5).

## 4. PeRQ

Both our deterministic and probabilistic bounds make clear that the worst-case outlier, and therefore the worst-case quantization error, is governed by blocks with the largest mass. In particular, the bound in Equation 2 implies

$$\|X\tilde{R}\|_\infty \leq \max_j \delta_{\{j\}} \sqrt{b}\|X_{\{j\}}\|_\infty \propto \max_j \|X_{\{j\}}\|_1,$$

which follows from the definition $\delta_{\{j\}} = \|X_{\{j\}}\|_1/(b\|X_{\{j\}}\|_\infty)$. Thus, for a fixed block size $b$, the tightest bound is achieved by minimizing $\max_j \|X_{\{j\}}\|_1$, which corresponds to balancing the per-block $\ell_1$ norms. Intuitively, for a fixed $X$, block rotations are most effective when large-magnitude coordinates are distributed evenly across blocks rather than concentrated within a subset of them. Empirically, we observe that the bound from Proposition 3.2 closely tracks actual per-token quantization error (Figure 5), extending Figures 3 and 4 from outlier suppression to quantization error itself. Together, this motivates our core design principle: permute activation coordinates to minimize the maximum per-block $\ell_1$ norm before applying block rotations.

**Overview.** PeRQ operationalizes the above design principle via the pipeline in Figure 2: given pre-trained weights and activations, we first compute a permutation that redistributes activation mass, then apply rotations, and finally round, optionally with error correction algorithms such as GPTQ (also known as OPTQ) (Frantar et al., 2023). Importantly, equalization is performed once on the full-precision model, and the resulting permutations are fused into surrounding layers before deployment by exploiting permutation equivariance within the compute graph (Definition 4.1).

**Mass Diffusion (MassDiff).** To balance activation mass across blocks, we propose MassDiff, a greedy permutation algorithm that directly targets the expected value of the maximum per-block $\ell_1$ norm over a calibration dataset (pseudocode in Algorithm 1). Figure 5 shows that this balancing directly translates to lower quantization error: MassDiff drives 77.2–100% of tokens to within 1% of the theoretical limit across our four models, reducing actual per-token quantization error by 37.5–40.5% on average over the identity

---

**Algorithm 1** Mass Diffusion (MassDiff) Algorithm

---

**Require:** Input activations from calibration dataset $\mathcal{D} = \{X^{(1)}, \ldots, X^{(m)}\}$, block size $b$, number of blocks $n = d/b$

$\quad \mathcal{J} \leftarrow \{1, \ldots, n\}$             ▷ *Initialize valid blocks*

$\quad \mathcal{B}_j \leftarrow \emptyset$ for all $j \in \mathcal{J}$         ▷ *Initialize per-block index sets*

$\quad \mathcal{I} \leftarrow \operatorname{argsort}_i(\frac{1}{m} \sum_{k=1}^{m} |X_i^{(k)}|)$     ▷ *Sort coordinates by descending average magnitude*

$\quad$ **for** $i \in \mathcal{I}$ **do**

$\quad\quad j^* = \arg\min_{j \in \mathcal{J}} \frac{1}{m} \sum_{k=1}^{m} \|X_{\mathcal{B}_j}^{(k)}\|_1 + |X_i^{(k)}|$    ▷ *Assign $i$ to minimize average maximum per-block $\ell_1$ norm*

$\quad\quad \mathcal{B}_{j^*} \leftarrow \mathcal{B}_{j^*} \cup \{i\}$          ▷ *Add $i$ to the selected block*

$\quad\quad$ **if** $|\mathcal{B}_{j^*}| = b$ **then**

$\quad\quad\quad \mathcal{J} \leftarrow \mathcal{J} \setminus \{j^*\}$         ▷ *Drop $j^*$ once the block is full*

$\quad\quad$ **end if**

$\quad$ **end for**

$\quad$ **Return** $[\mathcal{B}_1, \ldots, \mathcal{B}_n]$        ▷ *Concatenate blocks into the permutation*

---

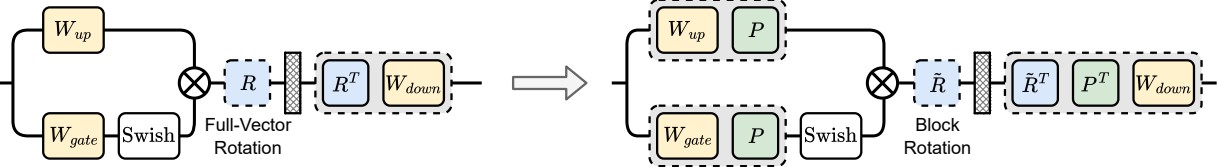

*Figure 6.* PeRQ replaces full-vector rotation $R$ with permutation $P$ and block rotation $\tilde{R}$, as shown here for the feedforward network in the standard transformer architecture. Importantly, this subgraph is a permutation-equivariant region (Definition 4.1) but it is not rotation-equivariant due to the Swish activation function. So, $P$ is merged into the surrounding weights, while $R$ and $\tilde{R}$ remain online.

permutation. By contrast, an alternative strategy is to assign coordinates to blocks in a zigzag pattern, as studied by Lin et al. (2024a) to reduce variance. This proxy only inadvertently tightens our bound, reaching the theoretical limit on 0.0–0.8% of tokens yet still reducing error by 21–36%. This reinforces our analysis on two fronts: (1) per-block $\ell_1$ balance is the right proxy, and (2) tightening the bound, even partially, translates to lower quantization error. This link carries through end-to-end: MassDiff improves both perplexity and zero-shot accuracy (Section 5), with further ablations across alternative permutation strategies and calibration scales in Appendix C.

**Permutation Equivariance in Neural Networks.** At first glance, introducing permutations may appear to alter the network computation. However, permutations can be introduced to preserve functional behavior without introducing overhead. We first define permutation-equivariant regions.

**Definition 4.1** (Permutation-Equivariant Region). A permutation-equivariant region is a contiguous subgraph of a neural network of the form $\Phi := f_n \circ \cdots \circ f_1$ whose map $\Phi : (\mathbb{R}^d)^m \to (\mathbb{R}^d)^{m'}$ satisfies the following property: for any activation vectors $X^{(1)}, \ldots, X^{(m)} \in \mathbb{R}^d$ and any permutation matrix $P \in \mathbb{R}^{d \times d}$ acting on the feature dimension, if $\Phi(X^{(1)}, \ldots, X^{(m)}) = (Y^{(1)}, \ldots, Y^{(m')})$ then $\Phi(X^{(1)}P, \ldots X^{(m)}P) = (Y^{(1)}P, \ldots, Y^{(m')}P)$.

Note that elementwise operations such as Swish or ReLU activation functions are equivariant to permutations along the feature dimension. Permutation-equivariant regions are therefore prevalent in neural network architectures, enabling one to introduce permutations without modifying the execution graph, as formalized in the following remark.

**Remark 4.2** (Merging Permutations). Within permutation-equivariant regions, one can freely commute permutations through the corresponding subgraph to be absorbed into surrounding linear layers. For example, consider activation $X \in \mathbb{R}^d$, permutation matrix $P \in \mathbb{R}^{d \times d}$, and weights $W_1, W_2 \in \mathbb{R}^{d \times d}$. Let permutation-equivariant region $\Phi : \mathbb{R}^d \to \mathbb{R}^d$ be unary (*i.e.*, $m = m' = 1$). Then,

$$\Phi(XW_1)W_2 = \Phi(XW_1P)P^TW_2 = \Phi(X\tilde{W}_1)\tilde{W}_2,$$

where $\tilde{W}_1 = W_1P$ and $\tilde{W}_2 = P^TW_2$. Therefore, for $g_i(X) = XW_i$, regions of the form $\Psi := g_2 \circ \Phi \circ g_1$ are functionally equivalent under permutations of the feature dimension. This property enables PeRQ to equalize activation mass across blocks with permutations that can be merged into surrounding weights before deployment, therefore leaving the graph unaltered and incurring no additional inference-time overhead. Figure 6 illustrates this process for the feedforward network of a standard transformer block, where the Swish activation function and elementwise multiplication form a multi-input, single-output region within which permutation matrix $P$ can be commuted.

# 5. Experimental Results

The core contribution of this work is PeRQ, our principled framework for improving the outlier suppression capabili-

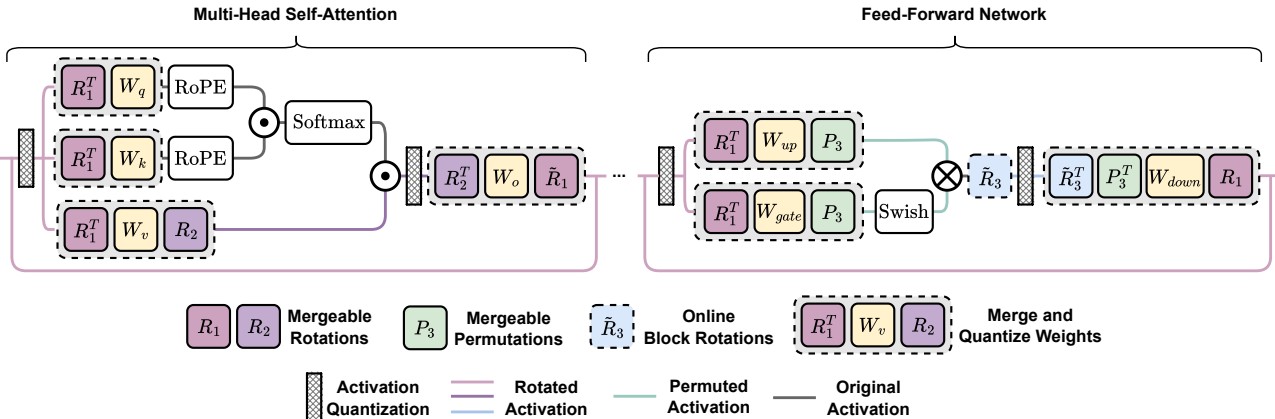

*Figure 7.* We illustrate our quantization graph architecture for a standard transformer block, merging rotations and permutations wherever possible and quantizing the weights and activations for all linear layers.

ties of block rotations by equalizing activation mass across blocks within permutation-equivariant regions of a compute graph. We therefore design our experiments to isolate the impact of permutations on preserving the accuracy of block-rotated LLMs quantized to narrow data formats, and to empirically validate our theoretical analysis.

**Models & Datasets.** We conduct experiments on Llama3 (Grattafiori et al., 2024) and Qwen3 (Yang et al., 2025) models. We use the implementations and instruction fine-tuned checkpoints made publicly available via Huggingface (Wolf et al., 2020) without modification. We construct our calibration dataset using 128 random sequences of 2048 tokens sampled from the WikiText2 (Merity et al., 2017) training dataset, and evaluate perplexity on the corresponding test split. While perplexity serves as a standard proxy for generative quality, it does not directly reflect downstream reasoning performance; we complement it with an analysis of zero-shot accuracy to assess generalization. Specifically, we use LightEval (Habib et al., 2023) to evaluate five reasoning tasks: ARC (challenge and easy) (Clark et al., 2018), HellaSwag (Zellers et al., 2019), PIQA (Bisk et al., 2020), and Winogrande (Sakaguchi et al., 2021). We provide ablations on the calibration source, calibration size, and model architecture in Appendix C. We note that comprehensive LLM evaluation remains an open problem, and we leave deeper analyses for future work.

**Implementation Details.** To describe our implementations, we decouple the *quantization graph* from the *quantization pipeline*. The former specifies where permutations and rotations are inserted into the compute graph, while the latter defines how different methods are composed throughout our experiments. Figure 7 illustrates the quantization graph architecture primarily considered in our experiments. Following QuaRot (Ashkboos et al., 2024) and SpinQuant (Liu et al., 2025), rotations are merged into linear layers when

possible ($R_1$ and $R_2$) and online rotations are only present at the inputs to the down projection layer in the feedforward network ($\tilde{R}_3$). PeRQ is defined at the pipeline level as a framework for equalizing permutation-equivariant regions; concrete methods correspond to different compositions of permutation, rotation, and rounding algorithms within the pipeline visualized in Figure 2. We primarily evaluate two PeRQ compositions. **PeRQ\*** composes our MassDiff permutation with full-vector Hadamard rotations at $R_1$ and $R_2$, block Hadamard rotations at $\tilde{R}_3$, and Qronos rounding. **PeRQ†** uses the same permutation and block rotation at $\tilde{R}_3$, but learns full-vector rotations $R_1$ and $R_2$ via Cayley optimization (Liu et al., 2025) before rounding to nearest (RTN). For all PeRQ variants, we apply block rotations only at $\tilde{R}_3$ and equalize only those regions in the graph; $R_1$ and $R_2$ are full-vector and are not equalized. To isolate MassDiff's contribution from the rest of the pipeline, we evaluate "No Permute" variants of both PeRQ\* and PeRQ† on perplexity and zero-shot tasks (Appendix C, Table 10). We provide hyperparameters and additional implementation details in Appendix B, and further ablations on alternative permutation algorithms, rotation merging strategy, and pipeline composition in Appendix C.

**Baseline Details.** We compare PeRQ against MR-GPTQ (Egiazarian et al., 2026) and BRQ (Shao et al., 2025), which are both recent methods that apply block rotations and perform rounding using GPTQ. To ensure a controlled comparison, we evaluate MR-GPTQ and BRQ using the same quantization graph architecture illustrated in Figure 7, merging rotations into linear layers when possible. MR-GPTQ and BRQ are therefore equivalent within our experiment design: both correspond to applying merged block rotations at $R_1$ and $R_2$, and online block rotations at $\tilde{R}_3$ with an identity permutation $P_3 = I_d$, followed by GPTQ rounding. We study the unfused, fully online rotation architecture originally proposed by Egiazarian et al. 2026 separately in

*Table 1.* We present the WikiText2 perplexity of Llama3 models quantized to INT4, comparing block rotations with and without PeRQ across block sizes. We provide full-vector rotations for reference, which is equivalent to QuaRot. All results use Qronos for error correction; additional results with RTN and GPTQ are provided in Table 9. BF16 results can be found in Table 2.

| | | 16 | 32 | 64 | 128 | 256 | 512 | 1024 | 2048 | Full |
|---|---|---|---|---|---|---|---|---|---|---|
| **1B** | No Permute | 35.8 | 25.4 | 23.5 | 20.4 | 18.9 | 17.5 | 16.6 | 16.4 | 16.1 |
| | PeRQ* | 18.2 | 16.9 | 16.6 | **15.9** | **15.9** | 16.1 | 16.1 | 16.1 | 16.1 |
| **3B** | No Permute | 18.9 | 15.6 | 14.4 | 14.4 | 14.0 | 13.8 | 13.2 | 13.4 | 12.6 |
| | PeRQ* | 14.2 | 13.3 | 12.8 | 12.6 | **12.4** | 12.6 | 12.6 | 12.6 | 12.6 |
| **8B** | No Permute | 32.0 | 12.8 | 9.5 | 8.8 | 8.4 | 8.2 | 8.2 | 8.1 | **7.9** |
| | PeRQ* | 9.5 | 8.5 | 8.1 | 8.0 | **7.9** | **7.9** | **7.9** | **7.9** | **7.9** |

an ablation (Appendix C, Figure 9, Table 11). In addition to MR-GPTQ/BRQ, we introduce two variants that isolate the effect of the rounding stage: MR-RTN and MR-Qronos, which replace GPTQ with RTN and Qronos, respectively, while applying the same block rotations. We also evaluate BRQ-Spin, which learns block rotations at $R_1, R_2 \in \mathbb{R}^{b \times b}$ via Cayley optimization before rounding with GPTQ. We provide additional implementation details in Appendix B.

**Evaluation of Block Size.** Table 1 reports WikiText2 perplexity comparing block rotations with and without permutations when quantizing weights and activations to INT4. We use Qronos as our rounding algorithm here, and report additional results with RTN in Appendix C (Table 5).

Two clear trends emerge. First, even in the absence of permutations, model quality generally improves as block size increases, with perplexity approaching that of full-vector rotations (equivalent to QuaRot with Qronos). This behavior is consistent with Equation 3, which predicts post-rotation outliers are reduced as $b \to d$ with diminishing returns. Second, PeRQ consistently improves perplexity across all models and block sizes, with particularly large gains at small block sizes. Moreover, PeRQ completely closes the gap with block sizes of 128 or greater. This demonstrates the effectiveness of permutations in mitigating the limitations of block rotations as identified in Proposition 3.2.

**Comparisons with Block Rotation Baselines.** Table 2 reports WikiText2 perplexity and average zero-shot accuracy for Llama3 and Qwen3 models across multiple data formats. All methods share the same quantization graph and differ only in their pipeline composition. Following the MR-GPTQ and BRQ proposals, all block rotations in this comparison use a fixed block size of 32.

Our results show that INT4 is a stringent stress test for block rotations. MR-RTN and MR-GPTQ/BRQ frequently exhibit severe perplexity degradation in this regime, even with gradient-based rotation optimization (BRQ-Spin). Replacing GPTQ with Qronos substantially improves MR-style baselines, but remains far from the full-precision baseline. In comparison, PeRQ consistently improves perplexity and

on average maintains over 91% of the full-precision zero-shot accuracy across both model families, with and without gradients. These trends support our theoretical analysis: while improved rounding can further reduce quantization error, PeRQ directly addresses the limitations of block rotations identified in Section 3.2. Interestingly, we observe that PeRQ† sometimes achieves lower perplexity than the full-precision baseline (*e.g.*, Qwen3 1.7B and 4B). In most cases, these gains are accompanied by modest reductions in zero-shot accuracy, suggesting mild overfitting to the calibration dataset. We note that this behavior is consistent with prior observations in which fine-tuning from a checkpoint can improve test accuracy despite quantization in the loop (Colbert et al., 2024; Liu et al., 2026).

The interaction between quantization pipeline composition and target data format yields additional insights. MR-GPTQ and BRQ were primarily proposed for and evaluated on microscaling (MX) formats, which inherently mitigate outliers via group-wise scaling. In contrast, INT4 and FP4 rely on per-channel scaling and are therefore more sensitive to outliers. Accordingly, consistent with the results from Egiazarian et al. 2026 and Shao et al. 2025, we observe that MR-style baselines perform substantially better on MXFP4 than on FP4 or INT4. Nevertheless, even for MXFP4, PeRQ† consistently yields the best perplexity across both model families, and on average maintains 95% of the full-precision zero-shot accuracy. Interestingly, while FP4 is generally more forgiving than INT4 for MR-style pipelines, we highlight that PeRQ consistently performs better on INT4 than on FP4, with PeRQ† consistently improving over PeRQ* across all formats.

## 6. Discussion & Conclusions

We present the first systematic, non-asymptotic analysis of outlier suppression with block Hadamard rotations. Our analysis yields sufficient conditions under which Hadamard rotations suppress outliers, and reveals that the effectiveness of block Hadamard rotations is fundamentally limited by the activation block with the largest mass. Guided by

*Table 2.* We report WikiText2 perplexity and average zero-shot accuracy for Llama3 and Qwen3 models quantized to various data formats using different pipeline compositions, all instantiated within the same graph architecture visualized in Figure 7. All variants of MR-GPTQ and BRQ apply block rotations without permutation, while PeRQ variants also apply permutation-based equalization prior to rotation.

| | | Llama3 | | | | | | Qwen3 | | | | | |
|---|---|---|---|---|---|---|---|---|---|---|---|---|---|
| | | WikiText2 (↓) | | | 0-shot (↑) | | | WikiText2 (↓) | | | 0-shot (↑) | | |
| Format | Methods | 1B | 3B | 8B | 1B | 3B | 8B | 1.7B | 4B | 8B | 1.7B | 4B | 8B |
| BF16 | - | 11.8 | 9.8 | 6.5 | 51.8 | 59.8 | 66.0 | 15.2 | 12.2 | 8.6 | 48.4 | 50.9 | 53.5 |
| **INT4** | MR-RTN | 3e3 | 1e4 | 9e3 | 34.9 | 34.8 | 34.7 | 9e2 | 8e2 | 1e2 | 35.2 | 34.8 | 36.2 |
| | MR-GPTQ / BRQ | 2e3 | 6e3 | 7e3 | 35.1 | 34.9 | 35.1 | 9e2 | 2e2 | 79.5 | 35.3 | 36.3 | 36.5 |
| | MR-Qronos | 41.8 | 84.5 | 1e2 | 38.7 | 36.9 | 35.5 | 41.8 | 35.8 | 24.3 | 38.1 | 41.1 | 41.3 |
| | BRQ-Spin | 1e3 | 5e3 | 5e3 | 34.6 | 34.4 | 34.8 | 5e2 | 2e2 | 56.2 | 35.4 | 35.6 | 36.3 |
| | PeRQ* | 16.9 | 13.3 | 8.5 | **47.7** | 54.2 | 60.3 | 18.9 | 14.9 | 10.8 | 44.0 | **47.0** | 49.9 |
| | PeRQ† | **15.9** | **10.9** | **8.4** | 46.8 | **56.1** | **60.9** | **13.6** | **11.1** | **9.5** | **45.8** | 46.7 | **50.7** |
| **FP4** | MR-RTN | 70.0 | 40.5 | 26.6 | 39.2 | 44.5 | 46.2 | 93.0 | 4e2 | 1e2 | 37.3 | 35.6 | 38.5 |
| | MR-GPTQ / BRQ | 43.2 | 27.0 | 18.9 | 42.6 | 47.0 | 50.7 | 60.0 | 2e2 | 93.0 | 38.0 | 37.5 | 39.2 |
| | MR-Qronos | 23.9 | 15.9 | 11.6 | 44.4 | **51.9** | 53.5 | 29.2 | 72.5 | 26.2 | 40.8 | 37.8 | 42.5 |
| | BRQ-Spin | 51.2 | 31.1 | 20.4 | 41.4 | 47.0 | 48.8 | 54.5 | 1e2 | 39.2 | 39.2 | 37.5 | 41.1 |
| | PeRQ* | 21.0 | 16.6 | 10.6 | 44.2 | 51.6 | 54.9 | 21.4 | 16.6 | 12.0 | 42.4 | **45.5** | 47.7 |
| | PeRQ† | **18.0** | **12.8** | **9.8** | **46.1** | 51.5 | **55.6** | **15.6** | **12.4** | **9.8** | **43.2** | 45.0 | **49.6** |
| **MXFP4** | MR-RTN | 18.5 | 15.2 | 8.2 | 47.8 | 52.1 | 61.5 | 23.1 | 23.5 | 11.2 | 43.0 | 47.8 | 50.3 |
| | MR-GPTQ / BRQ | 14.2 | 11.2 | 7.4 | 50.3 | 57.5 | 63.4 | 16.9 | 14.9 | 9.6 | **46.8** | 48.7 | 52.8 |
| | MR-Qronos | 14.0 | 11.6 | 7.4 | **51.0** | **57.6** | 62.4 | 16.9 | 14.9 | 9.6 | 45.5 | 48.6 | 51.4 |
| | BRQ-Spin | 14.9 | 13.2 | 7.8 | 48.5 | 53.6 | **63.8** | 19.1 | 12.6 | 9.6 | 44.9 | 49.0 | 51.6 |
| | PeRQ* | 14.2 | 11.4 | 7.4 | 48.3 | 57.5 | 63.1 | 17.1 | 14.0 | 9.9 | 46.0 | 48.7 | 50.6 |
| | PeRQ† | **13.2** | **9.6** | **7.2** | 49.1 | 57.2 | 63.0 | **11.8** | **9.3** | **7.8** | 46.7 | **50.5** | **54.2** |

these insights, we design PeRQ to equalize blockwise activation mass prior to rotation. Our experimental results closely align with our analysis: PeRQ consistently improves model quality with block Hadamard rotations across block sizes, data formats, and model families, which validates our central claim that equalizing mass across blocks mitigates the limitations we identified. Moreover, PeRQ outperforms existing block rotation baselines, namely MR-GPTQ and BRQ variants. Finally, from a systems perspective, the overhead introduced by PeRQ is negligible; MassDiff calibrates permutations in under two minutes for Llama3 8B, and the resulting matrices are fully merged into the graph before deployment to avoid adding inference overhead.

**Towards Choosing a Block Size.** Our analysis offers principled insight into the choice of block size $b$ through the lens of outlier suppression. Probabilistically, we prove that increasing $b$ reduces worst-case post-rotation outliers with high probability (Proposition 3.4), capturing the general trend observed in our experiments (Table 1 and Appendix C, Table 5). Deterministically, however, we prove that the worst-case bound itself grows with $b$ (Corollary 3.3), offering insight into the deviations where smaller block rotations occasionally match or outperform larger ones, as also reported by Egiazarian et al. (2026). Computationally, it is known that smaller $b$ is favorable: online block rotations cost $\mathcal{O}(d \log b)$ versus $\mathcal{O}(d \log d)$ for full-vector rotations

(Appendix A). Taken together, these results characterize the accuracy–compute trade-off that governs $b$; while we do not offer a Pareto-optimal block size, deriving one from first principles remains an exciting direction for future work.

**Limitations.** Our analysis is based on non-asymptotic bounds characterizing worst-case behavior for activation outlier suppression with block Hadamard rotations. Empirically, our results suggest this worst-case analysis yields an effective optimization target despite its inherent coarseness, likely because few-bit quantization error is dominated by activation outliers in LLMs (Ashkboos et al., 2024). In practice, substantial variation exists within these bounds, depending on the complex interaction between models, datasets, and quantization schemes, among other components. This limitation is most clearly reflected in MX formats such as MXFP4, which inherently mitigate outliers via group-wise scaling; in this regime, outlier suppression with rotations is less critical, and the performance gap between PeRQ and MR-style baselines narrows. These observations highlight that the benefits of permutation-based equalization are most pronounced in settings where outliers materially impact quantization error. Finally, our theoretical analysis does not explicitly account for weight outlier suppression or alternative rotation structures, such as the Givens matrices studied in Xu et al. 2025. We leave such extensions for future work.

## Impact Statement

This paper presents work whose goal is to advance the field of efficient machine learning. Reducing the precision of weights and activations is known to reduce the maximum attainable model quality. As such, quantization, like other lossy compression methods such as pruning and distillation, can shift model behavior in subtle ways. Therefore, it is important for compressed models to be evaluated before deployment. Furthermore, there are potential societal consequences of machine learning in general, none which we feel must be specifically highlighted here.

**Reproducibility Statement.** We integrate PeRQ into the open-source Brevitas quantization library (Franco et al., 2025b). Code and instructions for reproducing experiments can be found here: https://xilinx.github.io/brevitas/dev/papers/perq.html.

## Acknowledgements

We would like to thank Michaela Blott, Gabor Sines, Ralph Wittig, Syed Naim, Yonas Bedasso, and Max Keihn from AMD for their support.

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

# A. On the Compute Requirements of Hadamard Rotations

The analysis in Section 3.1 establishes, both theoretically and empirically, that full-vector Hadamard rotations can strongly suppress outliers. However, as discussed in Section 2, existing methods often require inserting online rotations into the compute graph to exploit these benefits, introducing inference overhead. Although optimized implementations of Hadamard rotations exist (Fino & Algazi, 1976; Dao, 2023), this overhead remains non-negligible (Ashkboos et al., 2024). An emerging response is to restrict full-vector Hadamard rotations to operate on smaller partitions of an activation vector via block Hadamard rotations (Shao et al., 2025; Egiazarian et al., 2026). The following remark summarizes the compute requirements of full-vector and block Hadamard rotations.

**Remark A.1** (Compute Requirements of Hadamard Rotations). Consider applying a Hadamard rotation to an activation vector of dimension $d = kt$, where $t$ is the largest factor of $d$ that is not a power of two and $k$ is the remaining power-of-2 component. When $t = 1$, then $d$ is a power of 2 and the Hadamard rotation can be implemented via a butterfly decomposition with $\mathcal{O}(d \log d)$ operations (Fino & Algazi, 1976). However, for $t > 1$, the standard butterfly decomposition does not directly apply. In Appendix A.1, we show that a non-power-of-2 Hadamard rotation can be reduced to complexity $\mathcal{O}(dt + d \log k)$. Consequently, for any $d$, full-vector online Hadamard rotations incur at least $\mathcal{O}(d \log d)$ compute cost, which becomes significant at the large activation dimensions encountered in state-of-the-art LLM architectures. This motivates restricting online rotations to operate on smaller, power-of-2 partitions as a general strategy for reducing inference-time computation. Accordingly, a block Hadamard rotation with block size $b$ applies independent Hadamard rotations within each block and has complexity $\mathcal{O}(d \log b)$, yielding substantial compute savings when $b \ll d$.

Table 3 reports the minimum compute operations required to rotate the input activations of down projection layers in various Llama3 and Qwen3 models with full-vector or block Hadamard rotations.[1] Across models, restricting rotations to smaller blocks substantially reduces the compute requirements of online rotations.

*Table 3.* We present the minimum compute operations (additions and subtractions) required to apply full-vector and block Hadamard rotations to the down projection layer inputs in Llama3 and Qwen3 models. For each model, we list the dimension $d$ of the input activations, its decomposition into power-of-2 and non-power-of-two components $k$ and $t$, respectively, and the operation counts for block sizes $b \in \{32, 128, 512\}$ relative to full-vector rotations.

| Model | Size | $d$ | $k$ | $t$ | 32 | 128 | 512 | Full |
|---|---|---|---|---|---|---|---|---|
| Llama3 | 1B/3B | 8192 | $2^{13}$ | 1 | 40960 (38%) | 57344 (54%) | 73728 (69%) | 106496 |
| | 8B | 14336 | $2^{11}$ | 7 | 71680 (28%) | 100352 (39%) | 129024 (50%) | 258048 |
| Qwen3 | 1.7B | 6144 | $2^{11}$ | 3 | 30720 (36%) | 43008 (50%) | 55296 (64%) | 86016 |
| | 4B | 9728 | $2^9$ | 19 | 48640 (18%) | 68096 (25%) | 87552 (32%) | 272384 |
| | 8B | 12288 | $2^{12}$ | 3 | 61440 (33%) | 86016 (47%) | 110592 (60%) | 184320 |

## A.1. An Optimized Non-Power-of-2 Hadamard Rotation

Let $X \in \mathbb{R}^d$ be an activation vector and $R \in \mathbb{R}^{d \times d}$ be a Hadamard matrix, with $d \geq 4$. If the dimension $d$ is a power of 2, the rotation of $X$ by $R$ can be fully decomposed into butterfly stages, resulting in $\mathcal{O}(d \log d)$ add/subtract operations. If $d$ is not a power of 2, it can be factored into a power of 2 and a non-power-of-2 component as $d = kt$, where $t$ is the largest non-power-of-2 factor of $d$ and $k$ is the power-of-2 remainder. Because the matrix $R$ is constructed from $k' = \log(k) - 2$ recursive applications of a $4t$-dimensional base matrix, the overall computation naturally splits into two parts: (1) $k'$ radix-2 butterfly stages, followed by (2) $2^{k'}$ instances of $4t$-dimensional rotations.

The $4t$-dimensional rotations themselves can be further optimized by first computing all sums and differences over every group of four adjacent inputs, and then performing a final stage that adds or subtracts $4t$ groups of $t$ elements according to a sign pattern determined by the definition of the $4t$-dimensional Hadamard matrix. Figure 8 illustrates this rotation structure for an activation vector in the Llama3 8B model with dimension 14336. This dimension factorizes as $512 \times 28$, resulting in 9 radix-2 butterfly stages and 512 independent 28-dimensional rotations. Each 28-dimensional rotation is further decomposed into three stages: the first two stages compute seven groups of four-input additions and subtractions, and the final stage performs 28 groups of seven-input add and subtract operations.

---

[1]The full-vector costs depend strongly on the largest non-power-of-2 factor of the activation dimension, which can lead to higher operation counts even for smaller vectors (*e.g.*, Qwen3 4B vs. Qwen3 8B).

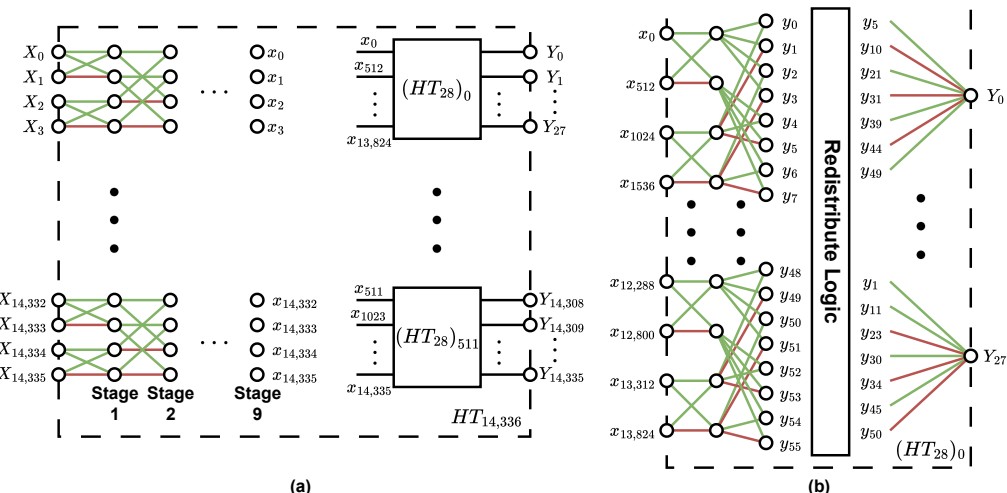

*Figure 8.* Structure of (a) 14336-dimensional Hadamard rotation and (b) 28-dimensional sub-rotation $(HT_{28})_0$. All lines marked in green are additions and the ones marked in red are subtractions.

Our proposed optimization reduces the number of add/subtract operations of a $d$-dimensional rotation from $\mathcal{O}(d(k'+4t-1))$ to $\mathcal{O}(d(k'+t+2))$, equivalently $\mathcal{O}(dt + d\log k)$. Due to the nature of the optimization, this provides the largest gains when $k'$ is small and $t$ is large. In particular, for fixed $k'$ and as $t \to \infty$, the reduction asymptotically yields a $4\times$ reduction in operation count. Table 4 reports the number of operations required for online rotations in non-power-of-2 dimensions found in Llama3 and Qwen3 models, comparing our reduction to existing techniques. Overall, our approach reduces the total number of operations by orders of magnitude relative to dense multiplication, and $1.4 - 2.9\times$ reduction in compute requirements compared to existing butterfly decomposition-based implementations (Dao, 2023).

*Table 4.* We report the number of operations required to rotate the input to the down projection layer in various LLMs with different methods, as well as the reduction in operations (in parentheses).

| Model | $d$ | $2^{k'} \times 4t$ | Matmul | Butterfly + Matmul | Ours |
|-------|-----|--------------------|--------|---------------------|------|
| Llama3-8B | 14336 | $2^9 \times 28$ | 205.51M $(796.4\times)$ | 516.10K $(2.0\times)$ | 258.05K |
| Qwen3-0.6B | 3072 | $2^8 \times 12$ | 9.43M $(236.3\times)$ | 58.37K $(1.5\times)$ | 39.94K |
| Qwen3-1.7B | 6144 | $2^9 \times 12$ | 37.74M $(438.8\times)$ | 122.88K $(1.4\times)$ | 86.02K |
| Qwen3-4B | 9728 | $2^7 \times 76$ | 94.62M $(347.4\times)$ | 797.70K $(2.9\times)$ | 272.38K |
| Qwen3-8B | 12288 | $2^{10} \times 12$ | 150.98M $(819.1\times)$ | 258.05K $(1.4\times)$ | 184.32K |

## B. Additional Implementation Details

**INT4 Quantization.** We use the $q$-bit integer quantizer defined as

$$\mathcal{Q}(X) = s \cdot \text{clip}\left(\left\lfloor \frac{X}{s} \right\rceil - z, \min \mathcal{A}, \max \mathcal{A}\right) + z, \tag{4}$$

where $s$ is the quantization step size (or scaling factor), $z$ is the zero-point, and $\lfloor \cdot \rceil$ is the round-to-nearest (RTN) operator (Zhang et al., 2022; Gholami et al., 2022). We quantize weights using the standard symmetric weight quantizer, where zero point $z = 0$ and scaling factor $s \in \mathbb{R}$ is optimized per-channel via linear search to minimize the mean squared error (MSE) loss between the full-precision and quantized weights, as is common practice (Ashkboos et al., 2024; Egiazarian et al., 2026; Zhang et al., 2026). Activations are quantized using the standard asymmetric activation quantizer, where zero point $z = \min(X)/s$ and scaling factor $s = (\max(X) - \min(X))/(2^q - 1)$ are dynamically defined per-token.

**(MX)FP4 Quantization.** We use the standard $q$-bit floating-point quantizer defined as

$$Q(X) := s \cdot \text{clip}\left(s'\left\lfloor \frac{X}{ss'} \right\rceil ; \min \mathcal{A}, \max \mathcal{A}\right), \tag{5}$$

where $\log s' = \max(\lfloor X/s \rfloor, 2^{e-1}) - m$ for target alphabet $\mathcal{A}$ defined by $m$ mantissa bits and $e$ exponent bits. We adhere to the OCP specification (Rouhani et al., 2023), which defines $e = 2$ and $m = 1$ for FP4. We quantize weights and activations symmetrically with zero point $z = 0$. For FP4 weights, scaling factor $s \in \mathbb{R}$ is optimized per-channel via linear search to minimize the MSE loss between the full-precision and quantized weights. For FP4 activations, scaling factor $s = \|X\|_\infty/(2^{q-1} - 1)$ is defined dynamically per-token. For MXFP4 weights and activations, scaling factors are defined for each group of 32 elements and are rounded down to a power of 2. For example, the scaling factor for the $j$-th block in activation vector $X$ is $\log s_{\{j\}} = \log \left( \lfloor \|X_{\{j\}}\|_\infty/(2^{q-1} - 1) \rfloor \right)$.

**Algorithm hyperparameters.** When applying GPTQ, the calibration set is constructed from 128 samples of 2048-token sequences drawn from the WikiText2 training set. We use the dampened covariance matrix $\tilde{X}^T \tilde{X} + \lambda I_d$, where $\tilde{X}$ denotes the rotated and quantized activations and $\lambda$ is 1% of the average diagonal of $\tilde{X}^T \tilde{X}$. We similarly use the dampened covariance matrix when applying Qronos, but choose $\lambda$ to be based on the maximum singular value of the covariance matrix such that $\lambda = \alpha \cdot \sigma_1$ with $\alpha = 1e^{-3}$ as proposed by Zhang et al. 2026. For both GPTQ and Qronos, we quantize weights in descending order of the diagonal of $\tilde{X}^T \tilde{X}$, as it has become common practice (Franco et al., 2025a; Colbert et al., 2025) and can provably improve error correction (Zhang et al., 2025). Note that, when composing PeRQ with GPTQ or Qronos, $\tilde{X}$ is also permuted. When calibrating permutations with MassDiff, we use only one sample of a 2048-token sequence randomly drawn from WikiText2; additional samples did not consistently yield additional benefit (see Appendix C, Table 7). When learning rotations, we draw 800 samples of 2048-token sequences for Cayley SGD, as proposed by Liu et al. 2025. In both cases, we still draw 128 samples for GPTQ or Qronos if error correction is applied. Unlike the original study by Liu et al. 2025, which optimized rotations after applying activation quantization but before weight quantization, we apply Cayley SGD for 100 epochs after both weights and activations have been quantized using the straight-through estimator (Bengio et al., 2013). All other hyperparameters (*e.g.*, learning rate and batch size) remain unchanged from the original study.

We implement all variants of PeRQ in PyTorch with the Brevitas quantization library based on v0.12.1 (Pappalardo et al., 2025), using their unmodified implementations of existing algorithms such as GPTQ, Qronos, QuaRot, and SpinQuant. We quantize all models using a single AMD Instinct MI210 GPU [2] with 64GB of memory.

## C. Ablations

We perform a series of ablation studies to assess the impact of key design choices, including the permutation method, rotation merging strategy, and pipeline composition. These experiments are intended to isolate the contribution of each component and to clarify how design decisions interact in practice.

**Isolating MassDiff with RTN.** Table 1 isolates permutation by comparing block rotations with and without MassDiff under a fixed Qronos rounding stage. Here, we further isolate MassDiff by also removing the choice of rounding algorithm: we replace Qronos with RTN, leaving only Hadamard rotations and the permutation itself. We quantize Llama3 1B, 3B, and 8B to INT4 (W4A4) and report WikiText2 perplexity in Table 5 across block sizes $b \in \{16, 32, 64, 128, 256, 512, 1024, 2048\}$ and full-vector rotations. MassDiff delivers orders-of-magnitude gains at small block sizes (e.g., Llama3 1B at $b = 32$ improves from 4e3 to 38.0, and 8B from 6e3 to 12.0), confirming that MassDiff permutations improve outlier suppression, independent of the rounding algorithm.

*Table 5.* We report WikiText2 perplexity ($\downarrow$) for Llama3 models quantized to INT4, comparing block rotations with and without MassDiff across block sizes, using RTN as the rounding algorithm. We provide full-vector rotations for reference.

|     |             | 16    | 32   | 64   | 128  | 256  | 512  | 1024 | 2048 | Full |
|-----|-------------|-------|------|------|------|------|------|------|------|------|
| 1B  | No Permute  | 5e3   | 4e3  | 2e3  | 6e2  | 2e2  | 58.0 | 27.0 | 25.8 | 38.0 |
|     | MassDiff    | 191.0 | 38.0 | 79.5 | 25.0 | 23.5 | **22.0** | 24.6 | 22.8 | 38.0 |
| 3B  | No Permute  | 2e4   | 7e3  | 9e3  | 1e3  | 2e2  | 54.5 | 21.8 | 20.1 | 22.0 |
|     | MassDiff    | 26.6  | 22.0 | 20.1 | 17.8 | **15.9** | 18.0 | 17.5 | 18.0 | 22.0 |
| 8B  | No Permute  | 2e4   | 6e3  | 2e3  | 3e2  | 93.0 | 42.5 | 16.4 | 10.9 | 12.0 |
|     | MassDiff    | 29.2  | 12.0 | 12.0 | 10.2 | 9.8  | **9.6** | 10.1 | 10.2 | 12.0 |

---

[2] AMD, AMD Instinct, and combinations thereof are trademarks of Advanced Micro Devices, Inc.

**Permutations.** In Table 6, we compare alternative permutation strategies within a fixed PeRQ pipeline using Qronos rounding and block rotations with $b = 32$ to quantize Llama3 and Qwen3 weights and activations to INT4. Random permutations are averaged over 5 seeds, "absmax" permutes coordinates in descending order of the maximum absolute value observed over the calibration sample, ZigZag uses the strategy proposed by Lin et al. 2024a, and MassDiff is given by Algorithm 1. Across model families and sizes, MassDiff consistently outperforms these alternative permutation methods, highlighting the importance of redistributing coordinates prior to rounding to balance activation mass.

*Table 6.* We report WikiText2 perplexity and average zero-shot accuracy for different permutation methods evaluated under a fixed PeRQ pipeline using block rotations with $b = 32$ and Qronos rounding, without learned rotations.

| | Llama3 | | | | | | Qwen3 | | | | | |
| | WikiText2 ($\downarrow$) | | | 0-shot ($\uparrow$) | | | WikiText2 ($\downarrow$) | | | 0-shot ($\uparrow$) | | |
| Method | 1B | 3B | 8B | 1B | 3B | 8B | 1.7B | 4B | 8B | 1.7B | 4B | 8B |
|---|---|---|---|---|---|---|---|---|---|---|---|---|
| No Permute | 25.6 | 15.6 | 12.9 | 43.8 | 48.6 | 35.4 | 43.2 | 16.6 | 12.6 | 38.2 | 46.7 | 48.0 |
| Random | 27.8 | 15.3 | 11.9 | 42.8 | 51.1 | 38.6 | 37.0 | 16.3 | 12.6 | 39.0 | 46.6 | 47.2 |
| Absmax | 20.1 | 14.4 | 10.4 | 45.7 | 54.6 | 58.6 | 25.4 | 15.9 | 11.8 | 42.9 | 47.6 | 48.5 |
| ZigZag | 17.1 | 13.4 | 8.6 | 46.6 | **54.8** | 58.1 | **18.9** | **14.9** | 10.9 | **44.0** | **47.6** | 49.6 |
| MassDiff | **16.9** | **13.3** | **8.5** | **47.7** | 54.2 | **60.3** | **18.9** | **14.9** | **10.8** | **44.0** | 47.0 | **49.9** |

**Permutation Calibration Size.** The comparisons in Table 6 calibrate each permutation with a single 2048-token sequence per permutation-equivariant region (Appendix B). Here, we revisit the comparison with substantially more data: we evaluate MassDiff, ZigZag, and a "No Permute" baseline on Llama3 1B quantized to INT4 with the PeRQ* configuration at block sizes $b \in \{16, 32, 64\}$, calibrating each permutation with 128K tokens per region. Table 7 reports WikiText2 perplexity: MassDiff matches or surpasses ZigZag at every block size, and both substantially improve over "No Permute". We hypothesize that this gap reflects the design of the two algorithms: MassDiff averages $\ell_1$ norms over the calibration set, so additional data may sharpen its estimate of blockwise mass per token, whereas ZigZag's coarse coordinate-wise $\ell_\infty$ reduction may benefit less from additional samples and could be sensitive to rare outliers.

*Table 7.* We report WikiText2 perplexity ($\downarrow$) for Llama3 1B quantized to INT4 with the PeRQ* configuration, comparing permutation strategies at block sizes 16, 32, and 64 with 128K calibration tokens per permutation-equivariant region.

| Permutation | b=16 | b=32 | b=64 |
|---|---|---|---|
| No Permute | 39.4 | 25.2 | 22.6 |
| ZigZag | 18.5 | 17.3 | 16.6 |
| MassDiff | **18.4** | **16.9** | **16.3** |

**Calibration Sensitivity.** Our main evaluation in Section 5 provides evidence of generalization when evaluating downstream reasoning: permutations calibrated on WikiText2 improve zero-shot accuracy across ARC (Challenge and Easy), PIQA, Winogrande, and HellaSwag on two model families and three model sizes (Table 2). Here, we assess sensitivity to the choice of calibration source itself. We evaluate Llama3 1B quantized to INT4 with the PeRQ* configuration (QuaRot rotations with Qronos rounding, $b = 32$), with and without MassDiff, across three calibration sources: C4 (Raffel et al., 2020), FineWeb (Penedo et al., 2024), and WikiText2 (Merity et al., 2017). We present the results in Table 8. Across all three sources, MassDiff consistently improves over the "No Permute" baseline in WikiText2 perplexity and on each of the five zero-shot tasks (note that the "No Permute" rows themselves vary across sources, since Qronos rounding also consumes the calibration data). The variation across calibration sources is modest and substantially smaller than the gain from MassDiff, indicating that MassDiff permutations generalize across calibration distributions as well.

**Pipeline Composition.** Table 9 shows how rounding choices in Stage 2 interact with different transformation strategies in Stage 1 within the PeRQ pipeline visualized in Figure 2. From the graph visualized in Figure 7, "MassDiff + QuaRot" corresponds to a pipeline where full-precision weights and activations are first permuted via MassDiff at $P_3$, then full-vector Hadamard rotations are applied at $R_1$ and $R_2$ and an online block Hadamard rotation is applied at $\tilde{R}_3$ to the permuted weights and activations. "MassDiff + SpinQuant" corresponds to a similar pipeline, but instead where full-vector rotations are learned at $R_1$ and $R_2$ via Cayley SGD. For these experiments, we use a block size of 32. Across pipelines, Qronos consistently outperforms GPTQ. However, when learning full-vector rotations, RTN consistently outperforms both. Thus, PeRQ* corresponds to "MassDiff + QuaRot + Qronos" while PeRQ† corresponds to "MassDiff + SpinQuant + RTN". These

*Table 8.* We report WikiText2 perplexity (↓) and zero-shot accuracy (↑) for Llama3 1B quantized to INT4 with the PeRQ* configuration, varying the calibration dataset used by both MassDiff and Qronos. Best result per calibration source is in bold.

| Calibration | Permutation | Wiki2 | ARC-C | ARC-E | PIQA | WinoG | HellaS | Avg |
|---|---|---|---|---|---|---|---|---|
| C4 | No Permute | 29.2 | 24.3 | 46.6 | 66.0 | 52.1 | 36.4 | 45.1 |
| | MassDiff | **18.9** | **28.9** | **53.8** | **68.0** | **52.2** | **39.1** | **48.4** |
| FineWeb | No Permute | 32.2 | 23.9 | 43.5 | 65.9 | 51.3 | 35.9 | 44.1 |
| | MassDiff | **18.8** | **26.5** | **51.0** | **67.0** | **52.2** | **38.4** | **47.0** |
| WikiText2 | No Permute | 25.4 | 24.4 | 46.0 | 64.0 | **53.1** | 35.9 | 44.7 |
| | MassDiff | **16.9** | **27.1** | **52.7** | **68.4** | 51.0 | **38.5** | **47.5** |

results highlight the complex interactions between rounding, rotation optimization, and permutation-based equalization. We leave a deeper investigation into this interaction for future work.

*Table 9.* We report WikiText2 perplexity and zero-shot accuracy for different compositions of Stage 1 (permutation and rotation) and Stage 2 (rounding) methods, following the pipeline illustrated in Figure 2.

| Stage 1 | Stage 2 | Llama3 | | | | | | Qwen3 | | | | | |
|---|---|---|---|---|---|---|---|---|---|---|---|---|---|
| | | WikiText2 (↓) | | | 0-shot (↑) | | | WikiText2 (↓) | | | 0-shot (↑) | | |
| | | 1B | 3B | 8B | 1B | 3B | 8B | 1.7B | 4B | 8B | 1.7B | 4B | 8B |
| MassDiff +QuaRot | RTN | 38.0 | 22.0 | 12.0 | 41.8 | 48.4 | 54.0 | 58.0 | 27.0 | 16.1 | 38.0 | 38.7 | 47.9 |
| | GPTQ | 28.4 | 14.9 | 9.8 | 44.9 | 52.5 | 58.9 | 25.8 | 21.0 | 12.6 | 43.4 | 37.9 | 49.3 |
| | Qronos | 16.9 | 13.3 | 8.5 | 47.7 | 54.2 | 60.3 | 18.9 | 14.9 | 10.8 | 44.0 | 47.0 | 49.9 |
| MassDiff +SpinQuant | RTN | 15.9 | 10.9 | 8.4 | 46.8 | 56.1 | 60.9 | 13.6 | 11.1 | 9.5 | 45.8 | 46.7 | 50.7 |
| | GPTQ | 19.5 | 13.4 | 9.2 | 46.2 | 55.5 | 60.5 | 20.8 | 17.5 | 12.8 | 42.5 | 41.4 | 47.5 |
| | Qronos | 16.4 | 13.2 | 8.5 | **48.4** | 55.9 | 59.7 | 18.2 | 14.7 | 10.6 | 44.5 | **47.4** | 50.2 |

**Isolating MassDiff via "No Permute" Baselines.** Within the PeRQ pipeline visualized in Figure 2, setting $P_3 = I_d$ in PeRQ* recovers MR-Qronos, and the analogous substitution in PeRQ† recovers a variant of SpinQuant.[3] We evaluate Llama3 8B quantized to INT4 in Table 10, reporting WikiText2 perplexity and the five zero-shot tasks; we also add GSM8K (Cobbe et al., 2021) as a reasoning-heavy generation task for this ablation, as it yields interesting new insights. Both PeRQ* and PeRQ† substantially outperform their "No Permute" counterparts on every task; the contrast is most pronounced on GSM8K, where MR-Qronos and the SpinQuant variant collapse to near-random while PeRQ* and PeRQ† recover much of the BF16 accuracy.

*Table 10.* We report WikiText2 perplexity (↓) and accuracy (↑) on five zero-shot tasks plus GSM8K for Llama3 8B quantized to INT4. MR-Qronos and SpinQuant respectively correspond to PeRQ* and PeRQ† with $P_3 = I_d$; the SpinQuant variant evaluated here uses an online block Hadamard rotation for $\tilde{R}_3$ with $b = 32$, matching our experimental setup.

| Method | Wiki2 | ARC-C | ARC-E | PIQA | WinoG | HellaS | GSM8K |
|---|---|---|---|---|---|---|---|
| BF16 | 6.5 | 49.7 | 77.4 | 80.5 | 64.6 | 57.5 | 75.4 |
| MR-Qronos | 1e2 | 21.2 | 27.9 | 54.0 | 50.1 | 26.5 | 1.7 |
| SpinQuant | 11.6 | 23.1 | 43.4 | 62.2 | 50.4 | 36.8 | 1.2 |
| PeRQ* | 8.5 | 40.6 | **70.2** | 72.5 | 58.5 | 51.2 | **57.3** |
| PeRQ† | **8.4** | **40.8** | 68.4 | **72.9** | **59.4** | 52.4 | 45.9 |

**Merging Rotations.** The quantization graph architecture originally proposed for MR-GPTQ by Egiazarian et al. 2026 exclusively applies online block Hadamard rotations (*i.e.*, no rotations are merged). In contrast, throughout our main experiments we study the quantization graph architecture illustrated in Figure 7, where rotations are merged into linear layers wherever possible in order to minimize changes to the deployed compute graph. Figure 9 illustrates the alternative graph architecture, in which all rotations are online block Hadamard rotations but where permutation-equivariant regions

---

[3]The original proposal by Liu et al. (2025) applies an online full-vector Hadamard at $R_3$ (*i.e.*, $b = d$), which we replace with a block Hadamard $\tilde{R}_3$ at $b = 32$ to match the rest of our experiments; see Appendix B for hyperparameter details.

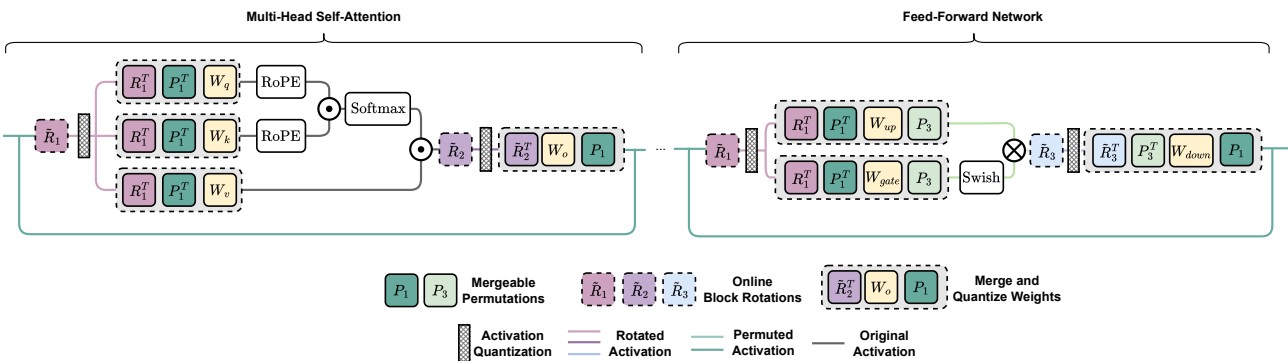

*Figure 9.* We provide an alternative quantization graph architecture where all rotations are left online, as proposed in (Egiazarian et al., 2026). We still merge permutations wherever possible and quantize the weights and activations for all linear layers.

can still be equalized and permutations can be merged prior to deployment. In this architecture, MR-GPTQ corresponds to $P_1 = P_3 = I_d$. To isolate the impact of this architectural choice, we evaluate both graph variants across data formats and report results in Table 11, where "merged" refers to the graph in Figure 7 and "online" refers to the graph in Figure 9. In both graph architectures, all block rotations use $b = 32$. Since learning full-vector rotations in the online setting would incur significant deployment overhead, we do not evaluate an online variant of PeRQ$^\dagger$ in this ablation. Nevertheless, PeRQ$^\dagger$ consistently yields the highest quality models.

*Table 11.* We report WikiText2 perplexity and average zero-shot accuracy across data formats for "merged" and "online" quantization graph architectures for MR-GPTQ and PeRQ.

| | | | Llama3 | | | | | | Qwen3 | | | | | |
| | | | WikiText2 ($\downarrow$) | | | 0-shot ($\uparrow$) | | | WikiText2 ($\downarrow$) | | | 0-shot ($\uparrow$) | | |
| Format | Method | Graph | 1B | 3B | 8B | 1B | 3B | 8B | 1.7B | 4B | 8B | 1.7B | 4B | 8B |
|---|---|---|---|---|---|---|---|---|---|---|---|---|---|---|
| **INT4** | MR-GPTQ | Merged | 2e3 | 6e3 | 7e3 | 35.1 | 34.9 | 35.1 | 9e2 | 2e2 | 79.5 | 35.3 | 36.3 | 36.5 |
| | | Online | 2e3 | 5e3 | 9e3 | 35.2 | 35.1 | 34.9 | 6e2 | 1e3 | 2e2 | 35.6 | 35.8 | 35.4 |
| | PeRQ* | Merged | 16.9 | 13.3 | 8.5 | **47.7** | 54.2 | 60.3 | 18.9 | 14.9 | 10.8 | 44.0 | **47.0** | 49.9 |
| | | Online | 26.6 | 13.4 | 9.1 | 39.5 | 51.7 | 53.7 | 20.4 | 17.5 | 12.8 | 44.0 | 45.8 | 46.7 |
| | PeRQ$^\dagger$ | Merged | **15.9** | **10.9** | **8.4** | 46.8 | **56.1** | **60.9** | **13.6** | **11.1** | **9.5** | **45.8** | 46.7 | **50.7** |
| **FP4** | MR-GPTQ | Merged | 43.2 | 27.0 | 18.9 | 42.6 | 47.0 | 50.7 | 60.0 | 2e2 | 93.0 | 38.0 | 37.5 | 39.2 |
| | | Online | 22.8 | 14.9 | 11.1 | 44.6 | 51.1 | **57.2** | 31.1 | 1e2 | 41.8 | 39.5 | 36.8 | 39.9 |
| | PeRQ* | Merged | 21.0 | 16.6 | 10.6 | 44.2 | 51.6 | 54.9 | 21.4 | 16.6 | 12.0 | 42.4 | **45.5** | 47.7 |
| | | Online | 23.5 | 16.1 | 10.4 | 44.8 | **52.2** | 55.0 | 22.4 | 17.8 | 14.0 | 41.8 | 45.2 | 47.4 |
| | PeRQ$^\dagger$ | Merged | **18.0** | **12.8** | **9.8** | **46.1** | 51.5 | 55.6 | **15.6** | **12.4** | **9.8** | **43.2** | 45.0 | **49.6** |
| **MXFP4** | MR-GPTQ | Merged | 14.2 | 11.2 | 7.4 | **50.3** | 57.5 | 63.4 | 16.9 | 14.9 | 9.6 | **46.8** | 48.7 | 52.8 |
| | | Online | 14.0 | 11.1 | 7.3 | 50.1 | **58.6** | **63.5** | 16.9 | 13.8 | 9.3 | 46.6 | 49.4 | 52.2 |
| | PeRQ* | Merged | 14.2 | 11.4 | 7.4 | 48.3 | 57.5 | 63.1 | 17.1 | 14.0 | 9.9 | 46.0 | 48.7 | 50.6 |
| | | Online | 14.0 | 11.6 | 7.4 | 49.2 | 58.0 | 62.6 | 17.1 | 14.0 | 9.6 | 45.8 | 48.8 | 51.7 |
| | PeRQ$^\dagger$ | Merged | **13.2** | **9.6** | **7.2** | 49.1 | 57.2 | 63.0 | **11.8** | **9.3** | **7.8** | 46.7 | **50.5** | **54.2** |

**Additional Experiments with SmolLM3.** The permutation equivariance property in Definition 4.1 holds for any contiguous subgraph of elementwise operations, which are prevalent across modern transformer architectures (Figure 7); PeRQ is therefore not architecture-specific. While our main experiments span Llama3 and Qwen3 across model sizes and dimensionalities (Table 2), here we add SmolLM3 3B as a third architecture and report INT4 W4A4 results in Table 12. Consistent with Table 2, PeRQ* and PeRQ$^\dagger$ substantially outperform MR-GPTQ and MR-Qronos on both WikiText2 perplexity and the five zero-shot tasks.

*Table 12.* We report WikiText2 perplexity (↓) and zero-shot accuracy (↑) for SmolLM3 3B quantized to INT4 (W4A4), following the same configuration as Table 2.

| Method | Wiki2 | ARC-C | ARC-E | PIQA | WinoG | HellaS |
|---|---|---|---|---|---|---|
| BF16 | 8.1 | 41.1 | 49.7 | 76.8 | 57.5 | 53.9 |
| MR-GPTQ | 2e3 | 21.1 | 26.5 | 54.6 | 50.1 | 26.6 |
| MR-Qronos | 15.6 | 28.7 | 43.2 | 66.5 | 52.8 | 38.9 |
| PeRQ* | **9.9** | **35.4** | **46.0** | **73.3** | **54.4** | **48.2** |
| PeRQ† | 10.0 | 33.3 | 45.8 | 71.8 | 53.5 | 46.1 |

# D. Proofs

## D.1. Proof of Proposition 3.1

**Proposition 3.1** (When Full-Vector Hadamard Rotations Suppress Outliers). *Let $X \in \mathbb{R}^d$ be an activation vector, and let $R \in \mathbb{R}^{d \times d}$ be a normalized Hadamard matrix. Define $\delta = \|X\|_1 / (d\|X\|_\infty)$, then the rotation of $X$ by $R$ satisfies*

$$\|XR\|_\infty \leq \delta\sqrt{d}\|X\|_\infty. \tag{1}$$

*Proof.* Given that $\|X\|_1 = \delta d\|X\|_\infty$, let $R_i \in \mathbb{R}^d$ denote the $i$-th column in $R$ for $i \in [d]$. Given that $R$ is a normalized Hadamard matrix, it follows that $\|R_i\|_\infty = 1/\sqrt{d}$. One can bound $|XR_i|$ via Hölder's inequality (Hölder, 1889) such that

$$|XR_i| \leq \|X\|_1 \|R_i\|_\infty,$$
$$|XR_i| \leq \delta\sqrt{d}\|X\|_\infty,$$

for all $i$. Thus, $\|XR\|_\infty \leq \delta\sqrt{d}\|X\|_\infty$, concluding the proof. □

**Remark D.1** (Outlier Suppression and Energy Concentration for Full-Vector Hadamard Rotations). We can arrive at a similar inequality as Equation 1 with an alternate definition of $\delta' = \|X\|_2/(\sqrt{d}\|X\|_\infty)$, which can be interpreted as the concentration of energy in $X$. Given that $R$ is a normalized Hadamard matrix, it follows that $\|R_i\|_2 = 1$ for $i \in [d]$. One similarly can bound $|XR_i|$ via Hölder's inequality (Hölder, 1889) such that

$$|XR_i| \leq \|X\|_2 \|R_i\|_2,$$
$$|XR_i| \leq \delta'\sqrt{d}\|X\|_\infty,$$

for all $i$. Thus, $\|XR\|_\infty \leq \delta'\sqrt{d}\|X\|_\infty$. Since $\|X\|_\infty \leq \|X\|_2 \leq \sqrt{d}\|X\|_\infty$, we have $\delta' \in [\sqrt{d^{-1}}, 1]$. This bound therefore states that only in the extreme case of $\delta' = 1/\sqrt{d}$ are rotations guaranteed to not make outliers worse (*i.e.*, $\|XR\|_\infty \leq \|X\|_\infty$). Therefore, unlike mass concentration, energy concentration does not deterministically yield a sufficient condition for outlier suppression with Hadamard rotations.

## D.2. Proof of Proposition 3.2

**Proposition 3.2** (When Block Hadamard Rotations Suppress Outliers). *Let $\tilde{R} = I_n \otimes R$ be a block rotation, where $R \in \mathbb{R}^{b \times b}$ is a normalized Hadamard matrix and $\tilde{R} \in \mathbb{R}^{d \times d}$ with $d = nb$. Let $X_{\{j\}} \in \mathbb{R}^b$ be the $j$-th block of $X \in \mathbb{R}^d$ for $j \in [n]$. Define $\delta_{\{j\}} = \|X_{\{j\}}\|_1 / (b\|X_{\{j\}}\|_\infty)$, then the rotation of $X$ by $\tilde{R}$ satisfies*

$$\|X\tilde{R}\|_\infty \leq \max_{j \in [n]} \delta_{\{j\}}\sqrt{b}\|X_{\{j\}}\|_\infty. \tag{2}$$

*Proof.* Given that block rotation $\tilde{R} = I_n \otimes R = \text{diag}(R, \ldots, R)$ is an orthonormal block-diagonal matrix composed of $n$ copies of a normalized Hadamard matrix $R \in \mathbb{R}^{b \times b}$, it follows that

$$\|X\tilde{R}\|_\infty = \max_j \|X_{\{j\}}R\|_\infty,$$

where $X_{\{j\}} \in \mathbb{R}^b$ is the $j$-th block of input activation vector $X \in \mathbb{R}^d$.

Given that $\|X_{\{j\}}\|_1 = \delta_{\{j\}} b\|X_{\{j\}}\|_\infty$, let $R_i \in \mathbb{R}^b$ denote the $i$-th column in $R$ with $i = 1, \ldots, b$. Given that $\|R_i\|_\infty = 1/\sqrt{b}$, one can follow Proposition 3.1 to bound $|X_{\{j\}} R_i|$ via Hölder's inequality (Hölder, 1889) such that

$$|X_{\{j\}} R_i| \le \|X_{\{j\}}\|_1 \|R_i\|_\infty,$$
$$|X_{\{j\}} R_i| \le \delta_{\{j\}} \sqrt{b} \|X_{\{j\}}\|_\infty,$$

for all $i = 1, \ldots, b$. Thus, $\|X\tilde{R}\|_\infty = \max_{j \in [n]} \|X_{\{j\}} R\|_\infty \le \max_{j \in [n]} \delta_{\{j\}} \sqrt{b} \|X_{\{j\}}\|_\infty$, concluding the proof. $\square$

**Remark D.2** (Outlier Suppression and Energy Concentration for Block Hadamard Rotations). Building from Remark D.1, let $\delta'_{\{j\}} = \|X_{\{j\}}\|_2 / (\sqrt{b} \|X_{\{j\}}\|_\infty)$. One can similarly bound $|X_{\{j\}} R_i|$ such that

$$|X_{\{j\}} R_i| \le \|X_{\{j\}}\|_2 \|R_i\|_2,$$
$$|X_{\{j\}} R_i| \le \delta'_{\{j\}} \sqrt{b} \|X_{\{j\}}\|_\infty,$$

for all $i = 1, \ldots, b$. Thus, $\|X\tilde{R}\|_\infty = \max_{j \in [n]} \|X_{\{j\}} R\|_\infty \le \max_{j \in [n]} \delta'_{\{j\}} \sqrt{b} \|X_{\{j\}}\|_\infty$, where $\delta'_{\{j\}} \in [\sqrt{b^{-1}}, 1]$ for all $j \in [n]$. This further implies that, unlike mass concentration, energy concentration does not deterministically yield a sufficient condition for outlier suppression with block Hadamard rotations.

### D.3. Proof of Corollary 3.3

**Corollary 3.3** (Deterministic Evolution of Post-Rotation Outliers). *Let $X \in \mathbb{R}^d$ be an activation vector and let $X_{\{j\}} \in \mathbb{R}^b$ be the $j$-th block of $X$ for $j \in [n]$ with $d = nb$. Define $\mathcal{Z}(b; X) = \max_{j \in [n]} \sqrt{b}\, \delta_{\{j\}} \|X_{\{j\}}\|_\infty$, where $\delta_{\{j\}} = \|X_{\{j\}}\|_1 / (b \|X_{\{j\}}\|_\infty)$. Then, for positive integers $k, b' \in \mathbb{N}$ such that $b = kb'$, it is verified that*

$$\mathcal{Z}(b; X) \le \sqrt{k}\, \mathcal{Z}(b'; X).$$

*Proof.* Given that $\|X_{\{j\}}\|_1 = \delta_{\{j\}} b \|X_{\{j\}}\|_\infty$, where $X_{\{j\}} \in \mathbb{R}^b$ denotes the $j$-th block of $X \in \mathbb{R}^d$ with $d = nb$, let $X_{\{j,i\}} \in \mathbb{R}^{b'}$ denote the $i$-th sub-block of $X_{\{j\}}$ for $i \in [k]$ with $b = kb'$. Then,

$$
\begin{aligned}
\mathcal{Z}(b; X) &= \max_{j \in [n]} \delta_{\{j\}} \sqrt{b}\, \|X_{\{j\}}\|_\infty = \max_{j \in [n]} \|X_{\{j\}}\|_1 / \sqrt{b} \\
&= \max_{j \in [n]} \left( \sum_{i \in [k]} \|X_{\{j,i\}}\|_1 \right) / \sqrt{kb'} \\
&\le \max_{j \in [n], i \in [k]} \sqrt{\frac{k}{b'}} \, \|X_{\{j,i\}}\|_1 = \sqrt{k}\, \mathcal{Z}(b'; X)
\end{aligned}
$$

Thus, $\mathcal{Z}(b; X) \le \sqrt{k}\, \mathcal{Z}(b'; X)$, concluding the proof. $\square$

### D.4. Proof of Proposition 3.4

We first state two auxiliary lemmas for completeness; both are well-known consequences of Hoeffding's lemma (Vershynin, 2018).

**Lemma D.3** (Rademacher Signed Sum is Sub-Gaussian, cf. Section 2.5 (Vershynin, 2018)). *Given $Y = (Y_1, \ldots, Y_d) \in \mathbb{R}^d$, let $S_1, \ldots, S_d$ be independent Rademacher random variables such that $S_i \sim \mathrm{Rad}(\pm 1)$, and define $Z = \sum_{i=1}^d S_i Y_i$. Then for all $t \in \mathbb{R}$,*

$$\mathbb{E}\big[e^{tZ} | Y\big] \le \exp\left( \frac{t^2 \|Y\|_2^2}{2} \right).$$

*In particular, conditional on $Y$, the random variable $Z$ is sub-Gaussian with variance proxy $\|Y\|_2^2$.*

**Lemma D.4** ($\ell_\infty$ Concentration from Conditional Sub-Gaussian, cf. Theorem 2.6.2 (Vershynin, 2018)). *Let $V = (V_1, \ldots, V_d) \in \mathbb{R}^d$ be a random vector and let $Y$ be a (possibly random) vector. Assume that, conditional on $Y$, each $V_i$ is zero-mean and sub-Gaussian with variance proxy $\nu_i^2(Y)$ in the sense that*

$$\mathbb{E}\big[e^{tV_i} | Y\big] \le \exp\left( \frac{t^2 \nu_i^2(Y)}{2} \right) \quad \text{for all } t \in \mathbb{R}.$$

*Then, conditional on $Y$, for all $\tau \geq 0$,*

$$\mathbb{P}(\|V\|_\infty \geq \tau \mid Y) \leq 2 \sum_{i=1}^{d} \exp\left(-\frac{\tau^2}{2\nu_i^2(Y)}\right) \leq 2d \exp\left(-\frac{\tau^2}{2\nu_{\max}^2(Y)}\right),$$

*where $\nu_{\max}^2(Y) = \max_{i \in [d]} \nu_i^2(Y)$.*

To prove Proposition 3.4, we separately consider the sign and magnitude of activation vector $X \in \mathbb{R}^d$ such that $X = \text{sign}(X) \odot |X| = S \odot Y$, then model $S = (S_1, \ldots, S_d)$ as a random vector with i.i.d. Rademacher entires, $S_i \sim \text{Rad}(\pm 1)$. As such, we clarify two core assumptions: (1) we assume $\mathbb{P}_S[+1 \mid Y] = \mathbb{P}_S[-1 \mid Y] = 0.5$, and (2) we assume each entry $S_i$ is i.i.d. Rademacher across vector coordinates within each activation vector. To ensure these assumptions are reasonable, we empirically evaluate Qwen3 1.7B using WikiText2. First, we find that the fraction of positive signs within a vector concentrates tightly around $0.5$, with a minimum of $0.47$, a max of $0.53$, and a mean of $0.50$ across tokens and layers. Second, we find that the pairwise correlations of $S_i$ for $i \in [d]$ are close to a Rademacher baseline estimated over 128 tokens; the off-diagonals of $\mathbb{E}_{S|Y}[S^T S]$ have a standard deviation that ranges between $0.08$ and $0.09$ across layers, where a Rademacher distribution would yield $1/\sqrt{128} = 0.088$. Thus, having verified our assumptions are reasonable, we proceed with proving Proposition 3.4 using Lemmas D.3 and D.4.

**Proposition 3.4** (Probabilistic Evolution of Post-Rotation Outliers). *Let $\tilde{R} = I_n \otimes R$ be a block rotation, where $R \in \mathbb{R}^{b \times b}$ is a normalized Hadamard matrix and $\tilde{R} \in \mathbb{R}^{d \times d}$ with $d = nb$. Given $Y = (Y_1, \ldots, Y_d) \in \mathbb{R}^d$, let $S = (S_1, \ldots, S_d)$ be a random vector with i.i.d. Rademacher entries $S_i \sim \text{Rad}(\pm 1)$. Define $X \in \mathbb{R}^d$ coordinate-wise as $X_i = S_i Y_i$ for $i \in [d]$. Then, conditional on $Y$,*

$$\|X\tilde{R}\|_\infty \leq \sqrt{\frac{2}{b} \log\left(\frac{2d}{\varepsilon}\right) \|X\|_2^2} \tag{3}$$

*with probability at least $1 - \varepsilon$.*

*Proof.* Let $\tilde{R} = I_n \otimes R$ be a block rotation, where $R \in \mathbb{R}^{b \times b}$ is a normalized Hadamard matrix with entries $R_{u,v} = \{\pm 1/\sqrt{b}\}$ for all $u, v \in [b]$, and $\tilde{R} \in \mathbb{R}^{d \times d}$ with $d = nb$. Given $Y = (Y_1, \ldots, Y_d) \in \mathbb{R}^d$, let $S = (S_1, \ldots, S_d)$ be a random vector with i.i.d. Rademacher entries $S_i \sim \text{Rad}(\pm 1)$. Define activation vector $X \in \mathbb{R}^d$ coordinate-wise as $X_i = S_i Y_i$ for $i \in [d]$; note that $|X_i| = |Y_i|$ for all $i$. Let $\tilde{X} = X\tilde{R}$ be the result of rotating $X$ by $\tilde{R}$.

Define block index $\beta(i) = \lceil i/b \rceil$ with corresponding index set $\mathcal{B}_{\beta(i)} = \{(\beta(i)-1)b+1, \ldots, \beta(i)b\}$. Since $\tilde{R} = I_n \otimes R$ with $R \in \mathbb{R}^{b \times b}$, the $i$-th coordinate of $\tilde{X}$ is then

$$\tilde{X}_i = \sum_{k \in [d]} S_k Y_k \tilde{R}_{k,i} = \sum_{k \in \mathcal{B}_{\beta(i)}} S_k Y_k \tilde{R}_{k,i}.$$

For $k \in \mathcal{B}_{\beta(i)}$, write $\tilde{R}_{k,i} = \frac{1}{\sqrt{b}} \gamma_{k,i}$ with $\gamma_{k,i} \in \{\pm 1\}$, and define $S_k^{(i)} = \gamma_{k,i} S_k$. Since $\gamma_{k,i}$ is fixed and $S_k \sim \text{Rad}(\pm 1)$, it follows that $S_k^{(i)}$ remain i.i.d. Rademacher for $k \in \mathcal{B}_{\beta(i)}$. Hence,

$$\tilde{X}_i = \frac{1}{\sqrt{b}} \sum_{k \in \mathcal{B}_{\beta(i)}} S_k^{(i)} Y_k.$$

Then, from Lemma D.3, it follows that, conditional on $Y$, $\tilde{X}_i$ is sub-Gaussian with variance proxy $\|Y_{\{\beta(i)\}}\|_2^2/b$, where $Y_{\{\beta(i)\}}$ denotes block $\beta(i)$ of $Y$ and $\beta(i) \in [n]$. In particular, since $|Y_k| = |X_k|$ for all $k \in \mathcal{B}_{\beta(i)}$,

$$\mathbb{E}[e^{t\tilde{X}_i} | Y] \leq \exp\left(\frac{t^2 \|X_{\{\beta(i)\}}\|_2^2}{2b}\right),$$

where $X_{\{\beta(i)\}}$ similarly denotes block $\beta(i)$ of $X$.

Since $S_i$ are zero-mean conditional on $Y$, it follows from Lemma D.4 that

$$\mathbb{P}\left(\|X\tilde{R}\|_\infty \geq \tau \mid Y\right) \leq 2d \exp\left(-\frac{b\tau^2}{2 \max_{j \in [n]} \|X_{\{j\}}\|_2^2}\right).$$

Solving for $\tau$ such that $\mathbb{P}\left(\|X\tilde{R}\|_\infty \geq \tau \mid Y\right) \leq \varepsilon$, it follows that

$$2d \, \exp\left(-\frac{b\tau^2}{2\max_{j\in[n]}\|X_{\{j\}}\|_2^2}\right) \leq \varepsilon \tag{6}$$

$$\frac{b\tau^2}{2\max_{j\in[n]}\|X_{\{j\}}\|_2^2} \geq \log\left(\frac{2d}{\varepsilon}\right)$$

$$\tau \geq \sqrt{\frac{2}{b}\log\left(\frac{2d}{\varepsilon}\right)\max_{j\in[n]}\|X_{\{j\}}\|_2^2}$$

Therefore, conditional on $Y$ and with

$$\tau = \sqrt{\frac{2}{b}\log\left(\frac{2d}{\varepsilon}\right)\max_{j\in[n]}\|X_{\{j\}}\|_2^2},$$

we obtain

$$\|X\tilde{R}\|_\infty \leq \sqrt{\frac{2}{b}\log\left(\frac{2d}{\varepsilon}\right)\max_{j\in[n]}\|X_{\{j\}}\|_2^2} \leq \sqrt{\frac{2}{b}\log\left(\frac{2d}{\varepsilon}\right)\|X\|_2^2}$$

with probability at least $1 - \varepsilon$, concluding the proof. $\square$

**Remark D.5** (Probabilistic Analysis of Outlier Suppression). Proposition 3.4 provides a complementary geometric perspective to the outlier suppression capabilities of block Hadamard rotations, namely that outlier suppression is probabilistically limited by the block with the largest mass. Recall the well-known relationship between mass and energy: for $X \in \mathbb{R}^d$, it can be shown that $\|X\|_2 \leq \|X\|_1$ via

$$\|X\|_1^2 = \left(\sum_{i\in[d]}|X_i|\right)^2 = \sum_{i\in[d]}X_i^2 + \sum_{i\in[d]}\sum_{j\in[d]\setminus i}|X_i||X_j| \geq \sum_{i\in[d]}X_i^2 = \|X\|_2^2.$$

It therefore follows from Proposition 3.4 that, conditional on $Y$, Equation 3 can be further bounded as

$$\|X\tilde{R}\|_\infty \leq \sqrt{\frac{2}{b}\log\left(\frac{2d}{\varepsilon}\right)\max_{j\in[n]}\|X_{\{j\}}\|_2^2}$$

$$\leq \sqrt{\frac{2}{b}\log\left(\frac{2d}{\varepsilon}\right)\max_{j\in[n]}\|X_{\{j\}}\|_1^2} = \sqrt{2\log\left(\frac{2d}{\varepsilon}\right)}\max_{j\in[n]}\delta_{\{j\}}\sqrt{b}\|X_{\{j\}}\|_\infty$$

with probability of at least $1 - \varepsilon$, where $\delta_{\{j\}} = \|X_{\{j\}}\|_1/(b\|X_{\{j\}}\|_\infty)$. Importantly, while this reduction again shows that outlier suppression is limited by the block with the largest mass, it does not yield a sufficient condition under which outlier suppression is guaranteed. Furthermore, building from Remarks D.1 and D.2, one can arrive at a similar insight using the energy concentration metric $\delta'_{\{j\}} = \|X_{\{j\}}\|_2/(\sqrt{b}\|X_{\{j\}}\|_\infty)$.

