# OpenReview forum: "Pushing the Limits of Block Rotations in Post-Training Quantization"
_ICML.cc/2026/Conference — ICML 2026 regular_

### Official Review · Reviewer_2cWJ · 2026-02-21

**Soundness:** 2
**Presentation:** 2
**Significance:** 3
**Originality:** 2
**Overall Recommendation:** 4
**Confidence:** 4

**Summary:**

The paper proposes MixQuant, a multi-bit quantization framework that enables extracting sub-models of different bit-widths from a single checkpoint. The method formulates a weighted multi-precision reconstruction objective and introduces a permutation-based mass balancing mechanism together with per-layer bit allocation for flexible deployment. Experiments across several LLM families show stable performance at low bit-widths.

**Compliance With Llm Reviewing Policy:**

Affirmed.

**Final Justification:**

Based on the authors’ rebuttal, my concerns regarding theoretical contribution and depth have been adequately addressed. The additional experiments isolating MassDiff’s effect and the evaluation on reasoning-heavy tasks (GSM8K) convincingly demonstrate the method’s practical utility. I therefore raise my rating to weak accept.

**Key Questions For Authors:**

1. On Generalizability. The core evaluation relies heavily on WikiText2 perplexity. While zero-shot accuracy is reported, these tasks may not stress long-form or reasoning-heavy generation. How does MixQuant perform on more challenging generative tasks that are sensitive to outlier-induced degradation, such as code generation (HumanEval) or mathematical reasoning (GSM8K)? Without evidence on such tasks, it's unclear whether the observed perplexity improvements translate to real-world utility.

2. On Theoretical Contribution. Proposition 3.1 and 3.2 provide upper bounds on post-rotation infinity norms using Hölder's inequality. While the interpretation of δ as mass concentration is intuitive, the analysis largely stops at this inequality. Could the authors provide a more profound theoretical principle (e.g., a necessary condition for outlier suppression? A more incisive answer would significantly raise the paper's theoretical merit and my evaluation.

3. Appendix B motivates block rotations by their lower inference cost (\(\mathcal{O}(d\log b)\) vs. \(\mathcal{O}(d\log d)\)). However, MixQuant adds calibration-time complexity via MassDiff. If the primary goal is inference-time efficiency, and given that learned full-vector rotations (SpinQuant) already achieve strong results (Table 6), why should a practitioner adopt the more complex block rotation + permutation pipeline over a simpler, better-performing full-vector learned rotation approach? Quantifying this trade-off more clearly would help position the work.

**Limitations:**

The authors have not adequately discussed the limitations of their work. It does not address: the theoretical limitations identified (the analysis being a direct application of Hölder's inequality); the potential overfitting of MassDiff permutations to the calibration set, etc.

**Strengths And Weaknesses:**

Strengths
* Addresses a practical deployment need: supporting multi-precision inference from a single quantized checkpoint.
* Formulates a unified multi-bit reconstruction framework that enables extracting sub-models at different bit-widths.
* Evaluated across multiple model families and bit configurations, with consistent empirical results.

Weaknesses
* While Propositions 3.1 and 3.2 provide upper bounds related to δ, they largely rely on Hölder-type norm inequalities and offer limited structural insight. The analysis does not explain why δ exhibits certain distributions in LLMs nor derive principled optimization strategies beyond the proposed greedy heuristic. The theory reads more as justification than as a design principle.

*  In Tables 1 and 6, the variant combining MassDiff with learned full-vector rotations (SpinQuant) significantly outperforms the variant using QuaRot + Qronos. This raises the question of how much of the observed gain is attributable to MassDiff itself. If most improvement stems from subsequent learned rotations, the standalone contribution of the permutation mechanism becomes less clear.

---

> ### Author Rebuttal · Authors · 2026-03-30
>
> Thank you for your review. We note that several elements of the summary and strengths do not appear in our paper (e.g., multi-bit framework). Below, we address the points that engage with our paper's content.
>
> **Re: Practical motivation, attribution, and generalization (Q1, Q3, W2, L2).**
> Thank you for these questions. To clarify, Table 1 was designed to isolate the benefits of MassDiff: No Permute and MixQuant* use the same Hadamard rotations with Qronos rounding, so the only difference is the permutation. Here, the gains are substantial: at $b$=16, Llama3 1B and 8B respectively improve from **35.8 to 18.2** and **32.0 to 9.5** on WikiText2.
>
> To address your question, we ran two sets of additional experiments with Llama3 8B quantized to INT4 to further isolate MassDiff's contribution. First, we complement Table 1 by replacing Qronos with RTN, leaving only Hadamard rotations:
>
> ||16|32|64|128|
> |-|-|-|-|-|
> |No Permute|2e4|6e3|2e3|3e2|
> |MassDiff|**29.2**|**12.0**|**12.0**|**10.2**|
>
> Second, we evaluate "No Permute" baselines for both MixQuant* and MixQuant†, i.e., MR-Qronos and SpinQuant, respectively.
>
> ||Wiki2|ARC-C|ARC-E|PIQA|WinoG|HellaS|GSM8K|
> |-|-|-|-|-|-|-|-|
> |BF16|6.5|49.7|77.4|80.5|64.6|57.5|75.4|
> |MR-Qronos|1e2|21.2|27.9|54.0|50.1|26.5|1.7|
> |SpinQuant|11.6|23.1|43.4|62.2|50.4|36.8|1.2|
> |MixQuant*|8.5|40.6|**70.2**|72.5|58.5|51.2|**57.3**|
> |MixQuant†|**8.4**|**40.8**|68.4|**72.9**|**59.4**|**52.4**|45.9|
>
> MixQuant* and MixQuant† each substantially outperform MR-Qronos and SpinQuant, both of which collapse to near-random on GSM8K as observed on other tasks (e.g., ARC-C). Thank you for encouraging the evaluation of reasoning-heavy generation as GSM8K reveals an even higher contrast. A broader study of reasoning-heavy generation under quantization is an interesting direction for future work, and these results serve as a step in that direction.
>
> To clarify, referring to Fig. 6, SpinQuant only trains mergeable full-vector rotations ($R_1$, $R_2$), leaving $\tilde{R}_3$ as an online block Hadamard rotation and $P_3$ as an identity matrix. MixQuant† instead calibrates $P_3$ using MassDiff to improve the outlier suppression capabilities of the block rotation; all else remain equal.
>
> Finally, Table 2 already provides evidence of cross-distribution generalization: permutations are calibrated on WikiText2, then evaluated on five distinct reasoning tasks across two model families and three model sizes. However, we ran additional experiments varying the calibration source itself and observe minor variation on zero shot accuracy (please see our response to Reviewer gyqw for more details). We will happily include these additional results in Appendix D.
>
>
> **Re: Theoretical contribution and depth (W1, Q2, L1).**
> Respectfully, we would appreciate further clarity as to what "a more profound theoretical principle" would entail, as we disagree with your characterization. Our work provides four distinct results: (1) Prop. 3.1 identifies the first non-asymptotic sufficient condition for full-vector outlier suppression; (2) Prop. 3.2 then provides the first generalization to block rotations; (3) Cor. 3.3 shows that increasing block size deterministically worsens this bound; and (4) Prop. 3.4 complements this by providing probabilistic evolution guarantees under light assumptions. Taken together, these results characterize the limits of block rotations and how they evolve with block size to yield the following geometric insight: **Both deterministically and probabilistically, outlier suppression is fundamentally limited by the mass concentration of the input vector.**
>
> Importantly, Fig. 3 shows that **$\delta$ is a remarkably predictive outlier suppression metric**; a strong correlation holds across both real LLM activations and synthetic distributions alike, showing that $\delta$ captures the geometry regardless of the underlying distribution. **MassDiff is a direct consequence of this analysis**: it minimizes the theoretical bottleneck identified by Prop. 3.2 (L237-247), and the isolation experiments we provided above further validate this predictive power end-to-end.
>
> **Our analysis makes the fewest assumptions of any prior works in this space.** The deterministic bounds require zero distributional assumptions and the probabilistic bound assumes only i.i.d. Rademacher signs (Appendix E.4). Since $\delta$ captures outlier suppression independent of any distribution, **avoiding such assumptions is a strength, not a limitation.** Prior work (e.g., Egiazarian et al.) assumes Gaussian or Laplacian distributions, which Fig. 3 shows are not predictive of real LLM activations.
>
> Finally, while we study the geometry of outlier suppression, characterizing the underlying geometry of LLM activation spaces (e.g., attention sinks, emergent outliers) remains an open problem. However, identifying $\delta$ as the operationally relevant metric is a critical first step before understanding why it exhibits certain distributions in LLMs.

---

> > ### Author Rebuttal · Reviewer_2cWJ · 2026-04-01
> >
> > After reading the reviews, my concerns on theoretical contribution and depth have been addressed. Therefore, I raise my rating for acceptance.

---

### Official Review · Reviewer_EgE7 · 2026-03-12

**Soundness:** 4
**Presentation:** 3
**Significance:** 3
**Originality:** 3
**Overall Recommendation:** 5
**Confidence:** 3

**Summary:**

The authors addressed the problem of outlier diffusion by block rotation used in post-training quantization methods. The authors conducted a thorough analysis of outlier suppression via block Hadamard rotations and proposed deterministic and probabilistic bounds for outlier suppression limits. Their work also introduced MixQuant, a rotation-aware PTQ method with permutation calibration. Their results showed improved quantized model accuracy for different block sizes. The permutation approach recovered up to 90% of full-vector rotation perplexity, compared to only 46% without permutation on quantized Llama3-1B. Overall, the work is well presented and the contributions are clear and meaningful.

**Compliance With Llm Reviewing Policy:**

Affirmed.

**Key Questions For Authors:**

1-	Regarding the probabilistic evolution of post-rotation outliers (Eq. 3): what guarantees the bound on post-rotation outliers, and how was it derived?

2-	A clear exposition and explanation of the utilization of the bounds for the permutation algorithm MaxDiff is needed, including parameter selection (e.g: suggested number of blocks). How are the valid indices initialized for each valid block?

3-	How does MixQuant apply other alternative permutation strategies? What is exactly meant by “other”? how is maxdiff utilized?

4-	Does permutation equivariance guarantee hold for other transformer layers? are there specific measures to guarantee constructing permutation matrix satisfying this equivariance property?

5-	What quantization pipeline does the MixQuant algorithm use, coupled with permutations? is it RTN/GPTQ?

**Limitations:**

Addressed

**Strengths And Weaknesses:**

Strengths:


1- The authors provided a comprehensive non-asymptotic analysis of outlier suppression for block rotation. They provided insightful bounds that account for the geometry of the input vectors. They  novelly provided necessary conditions and worst-case guarantees for outlier suppression via rotations.

2- Based on authors' derived limits, they developed a novel mass diffusion algorithm for constructing permutations to equalize outliers before rotation and quantization. They also showed that the permutation overhead can be minimized by merging permutations into surrounding layers in permutation-equivariant regions.

3- The theoretical analysis of outlier suppression is well presented and explained, the derived limits are justified by experimental results.

Weaknesses:


1- The MixQuant algorithm is not well explained and extremely summarized, the theoretical analysis is too detailed for main paper body. More space should be assigned for elaborating the MixQuant algorithm.

2- Hyperparameters regarding MixQuant initialization are not addressed

3- Experimental results should include other model families (e.g: Deepseek or Gemma) to ensure that the limits are generalizable across various architectures.

---

> ### Author Rebuttal · Authors · 2026-03-30
>
> Thank you for your thorough and constructive review. We address each of your points below.
>
> **Re: Algorithmic exposition (W1), hyperparameters (W2), block size selection (Q2), and alternative permutation strategies (Q3).** We appreciate the feedback on expanding the algorithmic exposition. We respectfully note that our theoretical insights are the contribution we are most excited to share. That said, the additional space afforded to a camera-ready revision would allow us to expand the algorithm discussion without reducing the theoretical content; such a revision could further clarify the initialization strategy for MassDiff, discuss the alternative permutation strategies, and consolidate block size selection guidance. We address these related points together. **First**, we will inline Appendix A into Section 4 and expand on the initialization strategy specified in Algorithm 1: blocks start empty, coordinates are sorted by descending average magnitude, and the greedy assignment proceeds deterministically by minimizing the maximum per-block $\ell_1$ norm (Appendix A, L560). **Second**, we will further clarify that MixQuant is a modular framework defined at the pipeline level (Section 5, L325-329): any permutation strategy can replace the "Permute" stage in Figure 2, and Table 5 (Appendix D) compares alternative strategies under identical conditions. **Third**, we will consolidate the theoretical and empirical guidance our paper provides on the choice of block size into Section 4. Proposition 3.4 shows that reducing block size can worsen post-rotation outliers with high probability, revealing a trade-off with the computational cost detailed in Remark B.1 (Appendix B). In Section 5, "Evaluation of Block Size" (L322-369), we complement this theoretical analysis with empirical results: MixQuant closes the gap to full-vector rotations at $b=128$ or larger. While we do not claim to derive the optimal block size, which is an exciting direction for future work, we will happily expand the exposition in Section 4 to centralize the theoretical and empirical guidance we do provide.
>
> **Re: The MixQuant quantization pipeline (Q5).** MixQuant is defined at the pipeline level (Section 5, L324-326), and is agnostic to the rounding algorithm: permutation and rotation are Stage 1 (transformation), while rounding is Stage 2 (Figure 2). MixQuant* uses Qronos and MixQuant† uses RTN. Table 6 (Appendix D) studies this decoupling, showing that MassDiff constructively composes with multiple rounding algorithms. We can further clarify this in a camera-ready.
>
> **Re: Additional model families (W3), and permutation equivariance (Q4).** Thank you for encouraging us to evaluate additional architectures. The permutation equivariance property (Definition 4.1) holds for any contiguous subgraph of elementwise operations, which are prevalent in standard transformer architectures (Figure 5). This means MixQuant naturally generalizes to additional model families (e.g., Gemma3 uses a GELUTanh instead of Swish, which is still permutation equivariant); the practical constraint is our current implementation's reliance on PyTorch Dynamo tracing to identify such regions programmatically, and we are actively working to resolve compatibility issues with Gemma3. Our experiments cover two model families (Llama3 and Qwen3) across model sizes with different dimensionalities (Table 3), and we now present additional results on SmolLM3 3B as another architecture. Below, we report W4A4 INT4 results corresponding to Table 2: WikiText2 perplexity (lower is better) and zero-shot accuracies (higher is better). Consistent with Table 2, MixQuant* and MixQuant† substantially outperform MR-GPTQ and MR-Qronos. We will be glad to include full results in a camera-ready revision.
>
> | Method | Wiki2 | ARC-C | ARC-E | PIQA | WinoG | HellaS |
> |------------|-------|-------|-------|------|-------|--------|
> | BF16 | 8.1 | 41.1 | 49.7 | 76.8 | 57.5 | 53.9 |
> | MR-GPTQ | 2e3 | 21.1 | 26.5 | 54.6 | 50.1 | 26.6 |
> | MR-Qronos | 15.6 | 28.7 | 43.2 | 66.5 | 52.8 | 38.9 |
> | MixQuant* | **9.9** | **35.4** | **46.0** | **73.3** | **54.4** | **48.2** |
> | MixQuant† | 10.0 | 33.3 | 45.8 | 71.8 | 53.5 | 46.1 |
>
> **Re: The probabilistic bound in Eq. 3 (Q1).** We provide the full proof in Appendix E.4, leveraging Lemma E.3 (Rademacher Signed Sum is Sub-Gaussian) and Lemma E.4 (L-infinity Concentration). The key assumption is that activation signs are i.i.d. Rademacher random variables, which we empirically validate on Qwen3 1.7B (Appendix E.4): intuitively, the fraction of positive signs concentrates around 0.5 across tokens and layers (min 0.47, max 0.53, mean 0.50). This is a lighter assumption than the Gaussian or Laplacian distributions assumed in prior work (e.g., Egiazarian et al. (2025)); Figure 3 shows prior distributional assumptions fail to capture the empirical behavior of $\delta$ in real LLM activations.

---

> > ### Author Rebuttal · Reviewer_EgE7 · 2026-04-01
> >
> > The authors addressed my concerns regarding the method hyperparameters and the permutation-equivariance guarantees for the studied architectures. However, I still believe the manuscript needs a clearer exposition of the theoretical insights and motivations behind the MixQuant design. I also remain concerned about the method’s generalizability under the strong assumption that the permutation-equivariance property holds. I understand that these concerns cannot be fully addressed within the short rebuttal period, and I believe my current score remains a fair assessment of the work quality. I therefore maintain the same rating.

---

### Official Review · Reviewer_UuEM · 2026-03-12

**Soundness:** 2
**Presentation:** 3
**Significance:** 3
**Originality:** 2
**Overall Recommendation:** 3
**Confidence:** 3

**Summary:**

This paper investigates the limitations of block Hadamard rotation in large-scale PTQ models. The authors point out that block rotation is less effective than full rotation in suppressing activation outliers due to the uneven distribution of activation mass among different blocks.

To address this, the paper proposes MixQuant, which redistributes channels via permutation before block rotation to ensure a more balanced ℓ1 mass for each block, thereby improving outlier suppression. Permutation is calculated during the calibration phase using a greedy algorithm (MassDiff) and can be incorporated into linear layers during inference, thus not increasing inference overhead. The paper also provides theoretical analysis and experimental verification on multiple LLMs.

**Compliance With Llm Reviewing Policy:**

Affirmed.

**Final Justification:**

I thank the authors for the detailed and thoughtful rebuttal. The additional clarifications on the differences from DuQuant, the expanded experimental results, and the discussion of the theoretical motivation are helpful and improve the overall clarity of the paper.

In particular, I find the paper technically sound and well-presented. The theoretical analysis provides useful intuition on the limitations of block-wise rotation, and the proposed approach is simple, practical, and well-integrated into existing PTQ pipelines. The zero-overhead design via permutation absorption is also a meaningful systems contribution.

However, my main concerns are only partially addressed by the rebuttal:

- **Originality**: While the distinction from DuQuant is clarified, the core idea—redistributing activation mass across blocks via permutation—remains conceptually close to existing outlier redistribution approaches. The contribution is practically useful but appears to be an incremental extension rather than a fundamentally new direction.

- **Scope and generality**: The method relies on permutation-equivariant regions and is primarily applicable to FFN blocks. This limits its applicability compared to more general approaches that can operate across different parts of the model (e.g., attention layers).

- **Theoretical contribution**: The analysis provides valuable intuition and aligns with empirical observations, but it remains largely explanatory or worst-case in nature. While the rebuttal shows encouraging correlations, the theory does not yet fully establish predictive power for real-world quantization error.

- **Evaluation**: The additional experiments are helpful, but the evaluation scope is still somewhat limited relative to the diversity of modern LLM benchmarks and settings.

Overall, I view this work as technically solid and practically relevant, especially from a deployment and systems perspective. However, due to the above concerns regarding novelty, generality, and the strength of the theoretical contribution, I remain somewhat unconvinced about its overall impact.

The rebuttal improves my understanding of the work and strengthens the empirical support, but does not fundamentally change my assessment. I therefore update my score to **3 (borderline)**.

**Key Questions For Authors:**

- **Q1**: What are the fundamental differences between this method and existing outlier redistribution/permutation-based methods (**DuQuant**)?

- **Q2**: Is there a significant correlation between the upper bound given by theoretical analysis and quantization error in the actual model?

- **Q3**: Does this method still offer significant gains in quantization formats with group scaling, such as MXFP4?

**Limitations:**

yes

**Strengths And Weaknesses:**

- **Strengths**
  - **Provides a clear theoretical analysis of the limitations of block rotation**

     The paper analyzes the upper bound of block rotation's ability to suppress outliers and points out that the imbalance of activation mass within a block is a significant cause of performance degradation.

   - **Intuitive insights and simple implementation**

     By redistributing activations through permutation, the mass of each block is made more balanced, thereby improving the performance of block rotation.

   -  **Reasonable engineering design**

      Permutation can be absorbed into the weights of linear layers, thus avoiding additional inference overhead.

- **Weaknesses**
  - **Limited Methodological Innovation**: The core idea is to redistribute activations among blocks through permutation, essentially a simple load-balancing strategy, similar to existing outlier redistribution approaches (**DuQuant**).

  - **Theoretical Results Primarily Worst-Case Bound**: Theoretical analysis primarily explains phenomena and has limited predictive power for actual quantization errors.

  -  **Limited Method Coverage**:  The analysis mainly targets activation outliers and does not consider weight distribution or other sources of quantization error.

  - **Narrow Experimental Evaluation Scope**: The analysis focuses mainly on perplexity and a limited number of tasks, lacking broader benchmark validation.

---

> ### Author Rebuttal · Authors · 2026-03-31
>
> Thank you for your review. We address each point below.
>
> **Re: DuQuant differentiation (W1, Q1).** This is an important distinction. To clarify, we do not claim permutations as a novel idea. Our contributions are three-fold: (1) the theoretical analysis identifying $\delta$ as a predictive outlier suppression metric, (2) the principled MassDiff algorithm that optimizes this metric, and (3) the zero-overhead system design enabled by permutation equivariance (Def. 4.1). MixQuant and DuQuant differ fundamentally in objective, yielding differences in:
> - **System design:** DuQuant is unable to merge permutations and thus requires $O(db)$ compute and $O(b^2)$ memory per layer (Sec. 2, L101-105); MixQuant adds *zero inference overhead* as the resulting permutations are fully merged into surrounding weights (Remark 4.2).
> - **Algorithms:** ZigZag (DuQuant) uses a coordinate-wise $\ell_\infty$ norm over a single calibration sample; MassDiff (MixQuant) minimizes $\ell_1$ norms averaged over a calibration set without coordinate-wise reduction.
> - **Theoretical analyses:** DuQuant's theoretical analysis targets variance reduction, while ours targets outlier suppression; we can elaborate on the precise technical distinctions in theoretical results if helpful.
>
> We highlight that more recent methods have adopted less expensive graph architectures (Sec. 2, L106-122). For example, as the most relevant concurrent work, Egiazarian et al. (i.e., MR-GPTQ) does not benchmark against DuQuant. However, while DuQuant's quantization graph architecture is incompatible with the ones we study (Figs. 6 and 8), we do study their permutation algorithm (i.e., ZigZag). Table 5 (App. D) compares MassDiff and ZigZag within the same pipeline, isolating the permutation strategy: MassDiff consistently matches or surpasses ZigZag across models.
>
> We further validate this with additional experiments increasing the calibration tokens to 128K per permutation-equivariant region. Below, we report WikiText2 perplexity of Llama3 1B quantized to INT4 for different block sizes:
>
> | |16|32|64|
> |-|-|-|-|
> |None|39.4|25.2|22.6|
> |ZigZag|18.5|17.3|16.6|
> |MassDiff|**18.4**|**16.9**|**16.3**|
>
> Importantly, MassDiff maintains its advantage as the calibration data scales, while ZigZag's coordinate-wise $\ell_\infty$ reduction loses cross-coordinate information (increasing data increases information loss for ZigZag).
>
> **Re: Predictive power of the theoretical analysis (W2, Q2).** Respectfully, we believe your characterization understates the predictive power demonstrated by our results; we present a logical deduction. Our analysis reveals that outlier suppression is fundamentally limited by the mass concentration of an input vector, namely $\delta$. Fig. 3 shows $\delta$ is strongly correlated with normalized post-rotation range, which governs worst-case quantization error (Sec. 3, L131-143). MassDiff is a direct consequence of this analysis, and Table 1 was designed to isolate the benefits of MassDiff: No Permute and MixQuant* use the same Hadamard rotations with Qronos rounding, so the only difference is the permutation. Thus, the substantial gains observed in Table 1 are directly attributed to the predictive power of our theoretical analysis.
>
> We ran additional experiments with Llama3 8B quantized to INT4 to further isolate this. The table below complements Table 1 by replacing Qronos with RTN:
>
> ||16|32|64|128|
> |-|-|-|-|-|
> |No Permute|2e4|6e3|2e3|3e2|
> |MassDiff|**29.2**|**12.0**|**12.0**|**10.2**|
>
> MassDiff reduces perplexity by orders of magnitude at small block sizes, validating that equalizing blockwise mass drives practical end-to-end gains and demonstrating that our theoretical analysis has strong predictive power on not just outlier suppression (Fig. 3), but also actual quantization errors.
>
> **Re: Evaluation scope (W4).** As a clarification, the existing evaluation in Table 2 evaluates zero-shot accuracy on 5 reasoning tasks across 2 model families and 3 model sizes (Sec. 5, L296-313). MixQuant improves normalized average accuracy consistently across all models. Since Table 2 reports normalized averages, the breadth of this evaluation may not be immediately apparent. We also evaluated GSM8K (see our response to Reviewer 2cWJ): MixQuant* achieves **57.3** on Llama3 8B INT4 while MR-Qronos collapsed (**1.7**). We will happily report individual task performance in App. D.
>
> **Re: Method coverage (W3).** We highlight that activation outliers are well-known to be the primary bottleneck in few-bit quantization; even without modeling weight distributions, our framework delivers consistent improvements (Tables 1 and 2). However, we have discussed this limitation in Sec. 6 (L435-438), and still believe such extensions are interesting for future work.
>
> **Re: MXFP4 (Q3).** We provide MXFP4 results in Table 2 and discuss limitations in Sec. 6 (L427-431). MixQuant still improves perplexity and zero-shot accuracy, but the benefit is most pronounced for INT4 and FP4.

---

> > ### Author Rebuttal · Reviewer_UuEM · 2026-04-03
> >
> > I thank the authors for their detailed and targeted rebuttal. I believe the authors have made effective clarifications and additions in the following aspects:
> >
> > - A clearer explanation of the differences between MassDiff and DuQuant in optimization objectives (mass vs. variance) and algorithm design;
> >
> > - Comparative experiments between MassDiff and ZigZag under a unified pipeline;
> >
> > - Added experiments under different calibration data scales and rounding methods, enhancing empirical support.
> >
> > These additions help to better understand the method's positioning and improve the paper's completeness.
> >
> > However, some of my core concerns have only received partial responses:
> >
> > 1. **Innovation:** Although the differences from DuQuant have been clarified, the method essentially still achieves the redistribution of activations among blocks through permutation, an idea conceptually similar to existing outlier redistribution methods. It is effective from an engineering perspective, but its novelty is still relatively limited.
> >
> > 2. **Scope of Application:** This method relies on a permutation-equivariant structure and primarily operates on FFN modules. While this design offers the advantage of zero inference overhead, it also limits its applicability to a wider range of modules (such as attention).
> >
> > 3. **Theoretical Analysis:** The theoretical section provides a good intuitive explanation, but it is largely a worst-case or interpretative analysis. Although experiments show some correlation, it is insufficient to prove its stable predictive ability for actual quantization errors.
> >
> > 4. **Experimental Scope:** Supplementary experiments are helpful, but the overall evaluation scope is still somewhat limited compared to current standards for LLM quantization work.
> >
> > Overall, I believe this work has some value in terms of engineering implementation and system design (especially zero-overhead deployment), but it still has some limitations in terms of methodological innovation and applicability.
> >
> > Based on the rebuttal, I am willing to adjust the rating to **3 (borderline)**.

---

> > > ### Author Response · Authors · 2026-04-04
> > >
> > > Thank you for the constructive engagement and for the updated score. We appreciate your acknowledgment of the DuQuant differentiation, the MassDiff vs. ZigZag comparison, and additional experiments. We hope the following responses address the residual points and we kindly ask that you further reconsider your assessment.
> > >
> > > **Re: Predictive ability of theoretical analysis.** Thank you for giving us another chance to address this. In our previous response, we used deductive reasoning to argue that our analysis is predictive: outliers dominate quantization error, so worst-case bounds target an impactful quantity. To further strengthen this argument, we directly compute the bound from Prop. 3.2 at $b$=32 and the actual quantization error, defined as $\Vert XP\tilde{R} - Q(XP\tilde{R})\Vert_F$, using a 4-bit quantizer for each token in Fig. 3. The resulting scatter plot (link below) reveals a tight correlation between our bound and actual quantization error (both normalized by $\Vert X\Vert_\infty$ per token). Compared pointwise to Identity ($P=I$), MassDiff reduces the bound for 100% of tokens across all four models, with mean quantization error reductions of 37.5-40.5%. In contrast, ZigZag only reduces the bound on 82-90% of tokens across models with 21-36% error reduction, confirming we identified the correct bottleneck: mass concentration. By directly operationalizing this insight, MassDiff consistently reduces the largest block-wise mass to the theoretical limit ($1/\sqrt{b}$) and, in doing so, reduces actual quantization error, confirming stable predictive ability. These new experiments isolate the theory and algorithm; our RTN isolation experiment and Table 1 show how this translates end-to-end.
> > >
> > > **Anonymized link to anonymized destination, compliant to guidelines**: [figure](https://github.com/user-attachments/assets/29c9253e-ef57-43c5-a7ae-17f873d193fc)
> > >
> > > **Re: Application scope.** To clarify, the FFN focus is a strategic priority, not a fundamental limitation. The down projection ($R_3$) has the largest online rotation cost and benefits most from block rotations: taking Llama3 8B as an example, at $b$=32, $R_3$ achieves 72% op reduction (258,048 to 71,680) while $R_1$ and $R_2$ achieve 58% and 29% reductions, respectively (see Table 3, Remark B.1 for more models). That said, we also note that Table 7 (App. D) demonstrates permutations applied to self-attention modules within the alternative graph architecture (Fig. 8). Finally, we clarify that permutation equivariance (Def. 4.1, Sec. 4, L272-284) is strictly less restrictive than rotation equivariance: permutations are a special case of orthogonal transforms, so any region admitting rotation merging also admits permutation merging. MixQuant exploits this to permute through the down projection, where permutation equivariance holds but general rotation equivariance does not (Fig. 5).
> > >
> > > **Re: Experimentation scope.** We appreciate the push to broaden our evaluation and would welcome specific suggestions for a camera-ready revision. Including our rebuttal experiments, our coverage now spans 7 models across 3 families (Llama3, Qwen3, SmolLM3), 3 data formats (INT4, FP4, MXFP4), and 6 reasoning tasks plus perplexity. For context, we surveyed 15 highly cited LLM quantization papers (e.g., GPTQ, SmoothQuant, QuaRot, SpinQuant) and selected the 5 zero-shot tasks with the highest adoption rates. We highlight that the new GSM8K results are particularly revealing: MixQuant* achieves 57.3 vs. MR-Qronos 1.7 on Llama3 8B INT4. If there are specific benchmarks or model families you consider essential, we are happy to prioritize them.
> > >
> > > **Re: Innovation and overall assessment.** We appreciate your acknowledgment that MixQuant has value in engineering and system design; however, our theoretical insights are the contributions that excite us the most. Specifically: (1) the theoretical discovery of mass concentration as a predictive metric for outlier suppression (Prop. 3.1, 3.2, Fig. 3) and actual quantization errors (Table 1, figure above), which no prior work identifies; (2) MassDiff, which operationalizes this theory to directly improve outlier suppression; and (3) the identification of permutation equivariance (Def. 4.1), enabling zero-overhead deployment. Our analysis bounds worst-case post-rotation outliers by design, because outliers dominate quantization error. Despite variation within these bounds, our worst-case granularity proves sufficient and actionable for quantization: our per-token experiments and end-to-end results confirm that tightening the bound consistently reduces actual quantization error. Thank you for pushing us on this point; your engagement directly strengthened our analysis and ultimately improves our paper. We believe the clarification and evidence presented address your residual concerns, and we would be grateful if you could reflect this in your assessment. We remain happy to incorporate any further revisions you identify.

---

### Official Review · Reviewer_gyqw · 2026-03-13

**Soundness:** 3
**Presentation:** 4
**Significance:** 4
**Originality:** 3
**Overall Recommendation:** 5
**Confidence:** 1

**Summary:**

This paper introduces MixQuant, a PTO framework that improves the accuracy by optimizing block-based rotations.
Previously people always rotating activation vectors to "hide" the activation outliers, but using small blocks usually is not that effective. It is largely do the the fact that the distribution of activation mass is not even across blocks.
MixQuant uses a greedy calibration algorithm Diffusion algorithm such that it will equalize the activation mass, which make the small block become more effective. Which provide important accuracy gain.

**Compliance With Llm Reviewing Policy:**

Affirmed.

**Final Justification:**

It is a well-written paper, I recommend it to be accepted.

**Key Questions For Authors:**

I want to know that how sensitive the MixQuant method to the dataset used for calibration? Is seems really important, will different dataset give different results? What if the calibration dataset is really different from dataset used in inference.

**Limitations:**

I want to understand how the sensitivity of the MixQuant method to the calibration dataset. It could be important as if the method really sensitive to the dataset, then I could get really different performance when my inference data is different from calibration dataset.

**Strengths And Weaknesses:**

MixQuant is a technically sound and highly practical framework that addresses the performance degradation of block-based rotations of PTO. Most of argument is well-supported by either math or empirically understanding.

The presentation is great and well structured. All the visualization makes sense.

Based on my limited understanding on this PTO field, I believe that the paper is both significant and mostly original.

---

> ### Author Rebuttal · Authors · 2026-03-30
>
> Thank you for your positive assessment and for highlighting the practical relevance and technical soundness of MixQuant. Your question about calibration sensitivity is well-taken.
>
> As a clarification, the existing evaluation provides substantial evidence of cross-distribution generalization: permutations are calibrated on WikiText2, but Table 2 evaluates zero-shot accuracy on five reasoning tasks (ARC-Challenge, ARC-Easy, PIQA, Winogrande, and HellaSwag) across two model families and three model sizes (Section 5, L296-313). MixQuant improves normalized average accuracy consistently across all models, indicating that calibrated permutations generalize well beyond the calibration source. However, since Table 2 reports normalized averages, the breadth of this evaluation may not be immediately apparent. We will happily report individual task performance in Appendix D in a camera-ready revision.
>
> To directly address your question, we also ran additional experiments varying the calibration source itself. Below we report W4A4 INT4 results for Llama3 1B (MixQuant* configuration from Table 1, with and without MassDiff) using three calibration datasets. We bold the best result per calibration dataset (lower is better for Wiki2 perplexity, higher is better for zero-shot accuracy).
>
> | Calibration | Permutation | Wiki2 | ARC-C | ARC-E | PIQA | WinoG | HellaS | Avg |
> |-------------|-------------|-------|-------|-------|------|-------|--------|-----|
> | C4 | No Permute | 29.2 | 24.3 | 46.6 | 66.0 | 52.1 | 36.4 | 45.1 |
> | C4 | MassDiff | **18.9** | **28.9** | **53.8** | **68.0** | **52.2** | **39.1** | **48.4** |
> | FineWeb | No Permute | 32.2 | 23.9 | 43.5 | 65.9 | 51.3 | 35.9 | 44.1 |
> | FineWeb | MassDiff | **18.8** | **26.5** | **51.0** | **67.0** | **52.2** | **38.4** | **47.0** |
> | WikiText2 | No Permute | 25.4 | 24.4 | 46.0 | 64.0 | 53.1 | 35.9 | 44.7 |
> | WikiText2 | MassDiff | **16.9** | **27.1** | **52.7** | **68.4** | **51.0** | **38.5** | **47.5** |
>
> We observe two trends. First, MassDiff consistently improves over the No Permute baseline regardless of calibration source, both in perplexity and across all five individual zero-shot tasks.
> Note that the No Permute rows also vary because Qronos rounding itself uses calibration data. Second, the variation across calibration datasets is modest; the improvement from MassDiff is substantially larger than the variation across calibration datasets.
>
> Thank you again for suggesting this experiment; it strengthens our analysis and ultimately our paper. We will be happy to include this calibration sensitivity analysis in a camera-ready revision. If these experiments answer your question, we would be grateful if you could reflect this in your confidence assessment.

---

> > ### Author Rebuttal · Reviewer_gyqw · 2026-04-04
> >
> > The rebuttal fully resolve my concerns. And I will maintain my positive score 5.

---

### Decision · Program_Chairs · 2026-04-30

**Decision:**

Accept (regular)

**Comment:**

Thanks for your submission. This submission proposes MixQuant, a framework to redistribute activaton mass via permutations prior to rotation to enable fewer post-rotation outliers for better quantization precision, backed up with rigorous analysis and practical algorithm. All reviewers and authors were actively engaged during the discussion phase. After the rebuttal, the majority reached the consensus of an acceptance recommendation. In the camera-ready version, please make sure to incorporate reviewer comments, including clarification of contribution in the context of other outlier redistribution approaches like DuQuant, improving the readability of the theoretical discussion, and incorporating more experimental results such as the SmolLM3 3B model and GSM8K evaluation.